# A Theoretical Study of Neural Network Expressive Power via Manifold Topology

**Jiachen Yao**                                                                  *jiacyao@cs.stonybrook.edu*
*Stony Brook University*

**Lingjie Yi**                                                                   *chris.yi@stonybrook.edu*
*Stony Brook University*

**Mayank Goswami**                                                               *mayank.isi@gmail.com*
*CUNY Queens College*

**Chao Chen**                                                                    *chao.chen.1@stonybrook.edu*
*Stony Brook University*

**Reviewed on OpenReview:** *https://openreview.net/forum?id=qRAjZuf48S*

## Abstract

A prevalent assumption regarding real-world data is that it lies on or close to a low-dimensional manifold. When deploying a neural network on data manifolds, the required size, i.e., the number of neurons of the network, heavily depends on the intricacy of the underlying latent manifold. While significant advancements have been made in understanding the geometric attributes of manifolds, it's essential to recognize that topology, too, is a fundamental characteristic of manifolds. In this study, we investigate network expressive power in terms of the latent data manifold. Integrating both topological and geometric facets of the data manifold, we present a size upper bound of ReLU neural networks.

## 1 Introduction

The expressive power of deep neural networks (DNNs) is believed to play a critical role in their astonishing performance. Despite a rapidly expanding literature, the theoretical understanding of such expressive power remains limited. The well-known *universal approximation theorems* (Hornik, 1989; Cybenko, 1989; Leshno et al., 1993; Hanin, 2017) guarantee that neural networks can approximate vast families of functions with an arbitrarily high accuracy. However, the theoretical upper bound of the size of such networks is rather pessimistic; it is exponential to the input space dimension. Indeed, these bounds tend to be loose, because the analyses are often oblivious to the intrinsic structure of the data. Real-world data such as images are believed to live in a manifold of a much lower dimension (Roweis & Saul, 2000; van der Maaten & Hinton, 2008; Jolliffe & Cadima, 2016). Such manifold's structure can be used to achieve better bounds of network size. It has been shown that the network size can be bounded by exponential of the manifold's intrinsic dimension rather than the encompassing input space dimension (Chen et al., 2019; Schmidt-Hieber, 2019).

However, the intrinsic dimension is only a small part of the manifold's property. It is natural to ask whether other properties of the manifold, such as topology and geometry, may lead to improved bounds. Safran & Shamir (2016) demonstrate that to approximate the indicator function of a $d$-dimensional ball, one only needs a network of size quadratic to $d$. However, this work assumes a rather simplistic input. To extend to a more general setting, one needs to incorporate the topology and geometry of the manifold into the analysis.

Early research has probed the geometry and topology of manifolds. Notably, Federer (1959); Amenta & Bern (1998) introduce a pivotal curvature measure, which adeptly captures the global geometric nuances of manifolds and has been embraced in manifold learning studies (Narayanan & Niyogi, 2009; Narayanan &

Mitter, 2010; Ramamurthy et al., 2019). On the topological front, descriptors like *Betti numbers* have been formalized in the language of algebraic topology to characterized the numbers of connected components and holes of a manifold (Hatcher, 2002; Bott et al., 1982; Munkres, 2018; Rieck et al., 2019). In their seminal work, Niyogi et al. (2008) integrate manifold's geometry and topology, setting forth conditions for topologically faithful reconstructions grounded in geometric metrics.

With the advent of the deep learning era, there has been a burgeoning interest in discerning the interplay between network size and manifold's intrinsic structural attributes. Existing studies (Dikkala et al., 2021; Schonsheck et al., 2019) bound network size with the geometry of manifolds. However, a theoretical framework that successfully integrates network size and topological traits has not yet been developed. This is a missed opportunity. The topological complexity of manifolds plays a crucial role in the learning problem, particularly concerning network size. Empirical findings (Guss & Salakhutdinov, 2018; Naitzat et al., 2020) suggest that even with similar geometry, data with larger topological complexity requires a larger network. These empirical observations highlight the need for a theoretical analysis that examines how manifold topology and geometry interact with network size. Addressing this need poses a significant challenge, as incorporating topological descriptors into the current analytical framework is inherently difficult due to the discrete nature of topology.

In this paper, we address this gap by presenting an innovative theoretical framework that integrates topology with neural network size. We tackle the challenge by first approximating a homeomorphism to disentangle the geometry and topology of a manifold. Notably, manifold topology remains invariant under a homeomorphism. This allows us to simplify the geometric complexity while maintaining the topological complexity. We then successfully represent discrete topological complexity as a combination of the complexities of basic topological shapes, thus overcoming the challenge posed by the discrete nature of topological descriptors. The resulting upper bound is obtained by constructing a neural network that first learns a low-dimensional embedding of the input manifold, followed by classification in the embedding space. This approach aligns with modern neural network design principles.

Our theoretical result reveals for the first time how the data topology, as a global structural characterization of data manifold, affects the network expressiveness. The beauty of the theorem is that it explicitly bounds the network size with both topology and geometry, giving us insights as to how the two important manifold properties affect learning. To capture the manifold's topology, we use the classic *Betti numbers*, which measure the number of connected components and holes within the manifold. For geometric measure, we use the *reach* introduced by Federer (1959); Amenta & Bern (1998), describing the manifold's overall flatness. See Figure 1 for illustrations of these measures.

Our main theoretical result is summarized informally below; the formal version is presented in Theorem 2.

**Main Theorem.** *(Informal) Let $\mathcal{M} \subset \mathbb{R}^D$ be a d-dimensional manifold ($d \leq D$) from a family of thickened 1-manifold and has two classes. There exists a ReLU network classifier $g$ with depth at most $O(\log \beta + \log \frac{1}{\tau})$ and size at most $O(\beta^2 + (\frac{1}{\tau})^{d^2/2})$, such that with large probability, the true risk of $g$ is small. $\beta$ is the sum of Betti numbers and $\frac{1}{\tau}$ is the inverse of reach.*

According to our bound, the network size scales quadratically in terms of the sum of Betti numbers $\beta$. Conversely, in terms of $\frac{1}{\tau}$, it scales as $O\left((\frac{1}{\tau})^{d^2/2}\right)$. This bound reveals interesting insights. The growth of network size is only affected by Betti numbers quadratically. Meanwhile, the network size can be affected by condition number more significantly when the intrinsic manifold dimension is high. These insights can be the foundation for future development of tighter bounds, and potentially inspire new designs of network architectures that capitalize on data's intrinsic manifold structures.

In the following section, we discuss related works and compare our derived bounds with those previously established. In Section 3, we define the problem and introduce the concept of the thickened 1-manifold family. At first glance, this family may appear overly restrictive; however, we will explain why it actually represents a broad and versatile class of manifolds. In Section 4, we present our theoretical findings, detailing the step-by-step derivation of our bounds. Our results establish a new theoretical perspective that can stimulate further exploration into the expressiveness of networks. Looking ahead, this theory could inform the design of more efficient neural networks by leveraging insights from manifold topology and geometry.

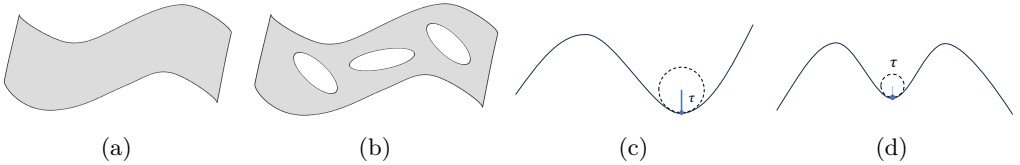

Figure 1: Illustration of *Betti numbers* and *reach*. (a) A 2-manifold embedding in $\mathbb{R}^3$ with $\beta_0 = 1, \beta_1 = 0$. (b) A 2-manifold embedded in $\mathbb{R}^3$ with $\beta_0 = 1, \beta_1 = 3$. (c) A 1-manifold with large reach. (d) A 1-manifold with small reach, which is the radius of the dashed circle.

## 2 Related Works

**Network size with manifold geometry.** Multiple studies have formulated network size bounds across varied manifold learning contexts based on geometry. Schonsheck et al. (2019) establish a bound of $O(LdD\epsilon^{-d-d^2/2}(-\log^{1+d/2}\epsilon))$ on the network size for manifold reconstruction tasks. $L$ is the covering number in terms of the injectivity radius, a geometric property. They utilize an auto-encoder, denoted as $D \circ E$, for the reconstruction of a manifold. Both the encoder $E$ and the decoder $D$ are designed to function as homeomorphisms. As a result, the overarching objective is the construction of a homeomorphism within the same space, which elucidates the absence of topological considerations in their outcomes. Our findings include not only the homeomorphism but also the classification network, with the latter being influenced by the manifold's topology. Chen et al. (2019) demonstrate the existence of a network of size $O(\epsilon^{-d/n}\log\frac{1}{\epsilon} + D\log\frac{1}{\epsilon} + D\log D)$ that can approximate any smooth real function supported on a compact Riemannian manifold. In this context, $n$ denotes the order of smoothness of the function. Their primary objective is to illustrate that, in manifold learning, the manifold dimension chiefly determines network size, with only a marginal dependence on the ambient dimension. Moreover, their smoothness assumption is inapplicable to classification tasks, where the target function lacks continuity. Yet, the interplay between manifold properties and their impact on network size in manifold classification largely remains unexplored.

**Classifier learned on manifold.** Dikkala et al. (2021) investigate network size in classification contexts. However, their foundational assumption is that a manifold's essence can be distilled into just two parameters: a centroid and a surrounding perturbation. They further assume there is a sensitive hashing property on manifolds. These assumptions are quite constrained, might not align with real-world complexities, and also overlooks the intrinsic properties of the manifold. Nevertheless, the aforementioned studies predominantly concentrate on network size and geometric traits, neglecting the equally critical role of topological features. Buchanan et al. (2021) establish a lower bound on the size of classifiers for inputs situated on manifolds. However, their theoretical framework is restricted to smooth, regular simple curves; it fails to account for complex manifold structures. Chung et al. (2018) focus on three types of manifolds: manifolds with strictly smooth convex hulls, manifolds of convex polytopes, and ring manifolds. They generate generic bounds on the manifold separability capacity using linear separation. Guss & Salakhutdinov (2018) provide empirical evidence showing that classifiers, when trained on data with higher Betti numbers, tend to have slower convergence rates. They also highlight that with rising topological complexity, smaller networks face challenges in effective learning. These findings underscore the need for a more comprehensive theoretical understanding.

There are some intriguing studies not primarily centered on manifold learning. Specifically, Bianchini & Scarselli (2014) establish a bound for the Betti number of a neural network's expression field based on its capacity. Nevertheless, their proposed bound is loose, and it exclusively addresses the regions a network can generate, neglecting any consideration of input manifold. Safran & Shamir (2016) explore the challenge of approximating the indicator function of a unit ball using a ReLU network. While their primary objective is to demonstrate that enhancing depth is more effective than expanding width, their approach has provided valuable insights. Naitzat et al. (2020) empirically examines the evolution of manifold topology as data traverses the layers of a proficiently trained neural network. We have adopted their concept of topological complexity. A number of studies, such as those by Hanin & Rolnick (2019) and Grigsby & Lindsey (2022), concentrate on exploring the potential expressivity of neural networks. However, these works primarily focus on the network's inherent capabilities without extensively considering the characteristics of the input data. Theoretically, Birdal et al. (2021); Andreeva et al. (2024) propose a persistent homology generalization

bound that accounts for the topology of the training trajectory. They propose a comprehensive metric that encompasses the neural network, optimization process, and network activations. In contrast, our work focuses specifically on approximating the indicator function of the data manifold using a neural network and studying the relationship between neural network size and the properties of the data manifold.

## 3    Preliminaries

**Topological Manifolds.**  In this paper, we focus on a specific class of manifolds called the *thickened 1-manifold family*, which are derived from operations on 1-manifolds. To lay the groundwork, we begin with the concept of a manifold. An n-dimensional manifold is a topological space where every point has a neighborhood that is homeomorphic (topologically equivalent) to $\mathbb{R}^n$. And an n-manifold with boundary allows for points that have neighborhoods homeomorphic to the half-space $\mathbb{R}_{\geq 0} \times \mathbb{R}^{n-1}$. For example, a circle $(s^1)$ is a 1-manifold without boundary, known as a closed manifold, while a line segment $(I^1)$ is a 1-manifold with boundary.

**Thickened 1-manifold.**  Our manifolds of interest are those obtained through operations on 1-manifolds, particularly when they are "thickened" to higher dimensions. Specifically, we consider *d-thickened 1-manifolds*, which are $(d+1)$-dimensional manifolds homeomorphic to $\mathcal{M}^1 \times B^d$, where $\mathcal{M}^1$ is a compact 1-manifold (with or without boundary), and $B^d$ denotes a *d*-dimensional closed ball. Figure 3 illustrates the 1-thickened 1-manifold. According to the classification theorem for 1-manifolds, $\mathcal{M}^1$ is homeomorphic to either a closed interval $I^1$ or a circle $S^1$. Consequently, a 1-thickened 1-manifold is homeomorphic to either $I^1 \times B^1$ or $S^1 \times B^1$.

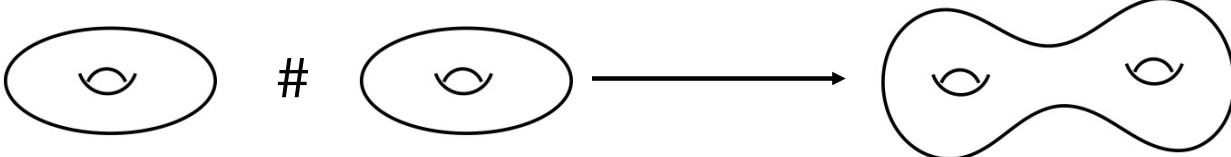

Figure 2: Illustration of connected sum.

**Connected Sum and Disjoint Union.**  To construct more complex manifolds from simpler ones, we utilize operations such as the connected sum and disjoint union. The connected sum of two closed manifolds $\mathcal{M}$ and $\mathcal{N}$, denoted $\mathcal{M} \# \mathcal{N}$, is formed by removing a small open ball from each manifold and then gluing them together along the resulting boundary sphere. The disjoint union $\mathcal{M} \sqcup \mathcal{N}$ combines two manifolds by considering them together without connecting them. These operations yield new compact manifolds that combine the topological features of the original manifolds. Although the connected sum of thickened 1-manifolds differs from the standard definition of the connected sum for closed manifolds mentioned earlier, we utilize the fact that the boundary $\partial \mathcal{M}$ of a thickened 1-manifold $\mathcal{M}$ is a closed manifold. The connected sum of two thickened 1-manifolds is defined by performing the connected sum on their boundaries and extend the gluing to the interiors of identifying corresponding points in the neighborhoods adjacent to the removed sphere. The formal definition of connected sum is provided in section A.1. Figure 2 illustrates the connected sum between thickened 1-manifolds.

With the tools defined above, we can now define the thickened 1-manifold family, which is the class of manifolds we will study in this paper.

**Definition 1** (Thickened 1-Manifold Family)**.**  *Let $\mathcal{M}^1$ represent a compact 1-manifold (with or without boundary). A d-thickened 1-manifold is a $(d+1)$-manifold homeomorphic to $\mathcal{M}^1 \times B^d$. The family of d-thickened 1-manifolds, denoted by $\mathbb{M}$, includes all such manifolds obtained via finite operations of connected sum and disjoint union.*

The proposed thickened 1-manifold family may appear straightforward at first. However, homeomorphisms, connected sums, and disjoint unions greatly enhance its expressivity, enabling a broad range of topological constructions. A homeomorphism allows the manifold to be smoothly deformed without altering its

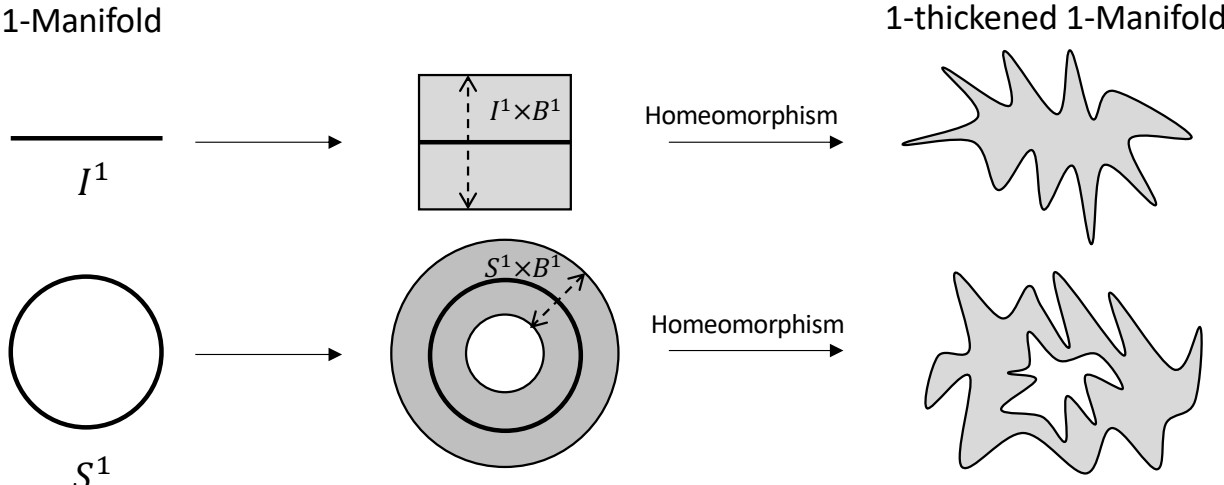

Figure 3: An illustration of the 1-thickened 1-manifold in 2D space. The top row shows a manifold that has the same homotopy type as a closed interval $I^1$, while the bottom row shows a manifold that is homotopy equivalent to a circle $S^1$. The 1-thickened 1-manifold family contains all manifolds that can be obtained through the disjoint union and connected sum of these manifolds.

fundamental topological properties, thus accommodating diverse geometric shapes. Connected sums can introduce additional "handles" or loops by merging two manifolds along shared boundaries, while disjoint unions allow multiple manifolds to coexist independently. These operations can yield manifolds of higher genus (e.g., multiple "holes") or several distinct loop-like components. As illustrated in Figure 3, even a single homeomorphism can significantly transform a manifold's embedded shape. Notably, all manifolds that can be contracted into skeletons belong to the thickened 1-manifold family. Examples include knots, multi-genus tori, and various high-dimensional "patches" formed through these constructions. Many real-world objects, such as cups, tires, and tables, also fall within this family due to their topological equivalence to thickened 1-dimensional structures. However, spheres in three-dimensional space are excluded, as thickened 1-manifolds only support non-trivial 0- and 1-dimensional Betti numbers, preventing the existence of higher-dimensional holes. While this assumption restricts the types of shapes considered, it also simplifies the computation of high-dimensional Betti numbers in real data. In fact, the concept of principal curves (Hastie & Stuetzle, 1989)—where data are assumed to cluster around a low-dimensional (1D) manifold—underscores the practicality of focusing on thickened 1-manifolds.

**Betti numbers.** We employ *Betti number* $\beta_k(\mathcal{M})$ to quantify topology of a manifold $\mathcal{M}$. $k$ is the dimension of that Betti number. 0-dimension Betti number $\beta_0(\mathcal{M})$ is the number of connected components in $\mathcal{M}$, and $\beta_k(\mathcal{M})$ ($k \geq 1$) can be informally described as the number of $k$-dimensional holes. 1-dimensional hole is a circle and 2-dimensional hole is a void. For the sake of coherence, we defer the formal definition of Betti numbers to Appendix A.1. Following Naitzat et al. (2020), we utilize the total Betti number of $\mathcal{M}$ as its topological complexity. The topological complexity we employ here is the topological complexity of data manifolds, instead of the topological complexity of neural networks used in Andreeva et al. (2024).

**Definition 2** (Topological Complexity). *$\mathcal{M}$ is a $d$-dimensional manifold. $\beta_k(\mathcal{M})$ is the $k$-dimensional Betti number of $\mathcal{M}$. The topological complexity is defined as*

$$\beta(\mathcal{M}) = \sum_{k=0}^{d-1} \beta_k(\mathcal{M}). \tag{1}$$

**Reach and condition number.** We then introduce metrics that encapsulate these geometric properties. For a compact manifold $\mathcal{M}$, the reach $\tau$ is the largest radius that the open normal bundle about $\mathcal{M}$ of radius $\tau$ is embedded in $\mathbb{R}^d$, i.e., no self-intersection.

**Definition 3** (Reach and Condition Number). *For a compact manifold $\mathcal{M} \subset \mathbb{R}^D$, let*

$$
\begin{aligned}
G = \{\mathbf{x} \in \mathbb{R}^D | \exists \mathbf{p}, \mathbf{q} \in \mathcal{M}, \mathbf{p} \neq \mathbf{q}, \\
||\mathbf{x} - \mathbf{p}|| = ||\mathbf{x} - \mathbf{q}|| = \inf_{\mathbf{y} \in \mathcal{M}} ||\mathbf{x} - \mathbf{y}||\}.
\end{aligned}
\tag{2}
$$

*The **reach** of $\mathcal{M}$ is defined as $\tau(\mathcal{M}) = \inf_{\mathbf{x} \in \mathcal{M}, \mathbf{y} \in G} ||\mathbf{x} - \mathbf{y}||$. The **condition number** $\frac{1}{\tau}$ is the inverse of the reach.*

Niyogi et al. (2008) prove that the condition number controls the curvature of the manifold at every point. A modest condition number $1/\tau$ signifies a well-conditioned manifold exhibiting low curvature.

**Problem setup.** In this paper, we examine the topology and geometry of manifolds in the classification setting. We have access to a training dataset $\{(\mathbf{x}_i, y_i) | \mathbf{x}_i \in \mathcal{M}, y_i \in [L]\}_{i=1}^n$, where $\mathcal{M} = \bigsqcup_{l=1}^L \mathcal{M}_l$. Each sample is drawn i.i.d. from a mixture distribution $\mu$ over $L$ disjoint manifolds with the corresponding label. For the simplification of notation, we build our theory on binary classification. It can be extended to multi-class without efforts in a one-verses-all setting. In binary case, the dataset is $\{(\mathbf{x}_i, y_i) | \mathbf{x}_i \in \mathcal{M}, y_i \in \{0, 1\}\}_{i=1}^n$, where $\mathcal{M} = \mathcal{M}_1 \sqcup \mathcal{M}_0 \in \mathbb{R}^D$. $\mathcal{M}_1$ and $\mathcal{M}_0$ are two disjoint $d$-dimensional manifolds representing two classes. The label $y_i$ is determined by the indicator function

$$
I_{\mathcal{M}_1}(\mathbf{x}) = \begin{cases} 1, & \mathbf{x} \in \mathcal{M}_1, \\ 0, & \text{otherwise.} \end{cases}
\tag{3}
$$

A neural network $h(\mathbf{x}) : \mathbb{R}^D \to [0, 1]$ approaches the classification problem by approximating the indicator function $I_{\mathcal{M}_1}(\mathbf{x})$. In the scope of this study, we focus on neural networks utilizing the ReLU (Rectified Linear Unit) activation function.

**Definition 4** (Adapted from Arora et al. (2018)). *A ReLU multi-layer feed-forward network $h : \mathbb{R}^{w_0} \to \mathbb{R}^{w_{k+1}}$ with $k + 1$ layers is defined as*

$$
h(\mathbf{x}) = h_{k+1} \circ h_k \circ \cdots \circ h_1(\mathbf{x}),
\tag{4}
$$

*where $h_i : \mathbb{R}^{w_{i-1}} \to \mathbb{R}^{w_i}, h_i(\mathbf{x}) = \sigma(W_i \mathbf{x} + b_i)$ for $1 \leq i \leq k$ are ReLU layers, and $h_{k+1} : \mathbb{R}^{w_k} \to \mathbb{R}^{w_{k+1}}, h_{k+1}(\mathbf{x}) = W_{k+1} \mathbf{x} + b_{k+1}$ is a linear layer. $\sigma$ is the ReLU activation function. The **depth** of the ReLU network is defined as $k + 1$. The **width** of the ReLU network is $\max\{w_1, \ldots, w_k\}$. The **size** of the ReLU network is $\sum_{i=1}^k w_i$.*

The approximation error of a ReLU network is determined by the true risk.

**Definition 5** (Approximation Error). *Let's consider the indicator function $I_{\mathcal{M}_1}$ for a manifold $\mathcal{M}_1$ in a binary classification problem where $\mathcal{M} = \mathcal{M}_1 \sqcup \mathcal{M}_0$. A neural network operates as a function $h(\mathbf{x}) : \mathcal{M} \to \mathbb{R}$. The approximation error of the neural network $h$ is then defined as:*

$$
\text{True Risk: } R(h) = \int_{\mathcal{M}} (h - I_{\mathcal{M}_1})^2 \mu(\mathbf{x}) d\mathbf{x}.
\tag{5}
$$

*$\mu$ is any continuous distribution over $\mathcal{M}$.*

## 4 Main Results

In this section, we explore how the topology of manifolds influence network size in classification scenarios. Our results, derived methodically through construction, follow two steps. First, we approximate a homeomorphism between the input manifold and a latent one; second, we carry out classification within this latent manifold. This latent manifold is designed to have simple geometric features, akin to those found in spheres and tori, while retaining the intrinsic topological characteristics of the original manifold. By design, the first phase is purely geometric, as the topological traits remain unaltered, while the subsequent classification phase is predominantly topological. Consequently, the required network size can be delineated into two distinct parts. We employ *Betti numbers* and the *condition number* as metrics to gauge topological and geometric complexities, respectively. Specifically, Betti numbers quantify the number of connected components and holes within the manifold, whereas the condition number characterizes the manifold's overall curvature.

### 4.1 Complexity Arising from Topology

To focus on the topological aspects rather than the geometric intricacies of the manifold, our attention shifts to the elemental shapes that typify the thickened 1-manifold collection. Among these shapes are the $d$-dimensional balls, denoted as $B_r^d(\mathbf{c})$, and the solid $d$-tori, represented by $T_{r,R}^d$. The $d$-dimensional ball, $B_r^d(\mathbf{c})$, is characterized by a radius $r$ and is centered at $\mathbf{c}$, mathematically defined as

$$B_r^d(\mathbf{c}) = \left\{ \mathbf{x} \in \mathbb{R}^d : \|\mathbf{x} - \mathbf{c}\|_2 \leq r \right\}. \tag{6}$$

On the other hand, the solid $d$-torus, $T_{r,R}^d$, embodies a genus-1 torus with a tunnel radius of $r$ and a tunnel center radius of $R$, centered at $\mathbf{c}$. Its formula is given by

$$T_{r,R}^d(\mathbf{c}) = \left\{ \mathbf{x} \in \mathbb{R}^d : (x_d - c_d)^2 + \left( \sqrt{\sum_{i=1}^{d-1} (x_i - c_i)^2} - R \right)^2 \leq r^2 \right\}. \tag{7}$$

It's pertinent to acknowledge that the general structure of a $d$-torus becomes significantly more complex as $d$ increases. The provided equation for $T_{r,R}^d$ represents only one of its potential configurations. Nevertheless, this depiction suffices within the context of the thickened 1-manifold ensemble.

**Lemma 1** (Topological Representative). *Let $\mathcal{M} \subset \mathbb{M}$ be a $d$-dimensional manifold from the thickened 1-manifold family. There exist a set of $m_1$ $d$-balls $\mathcal{B} = \{B_{r_i}^d(\mathbf{c}_i)\}_{i=1}^{m_1}$ and a set of $m_2$ solid $d$-tori $\mathcal{T} = \{T_{r_i,R_i}^d(\mathbf{c}_i)\}_{i=1}^{m_2}$, such that $\mathcal{M}$ is homeomorphic to the union $(\bigcup_{B \in \mathcal{B}} B) \cup (\bigcup_{T \in \mathcal{T}} T)$, where $m_1 + m_2 \leq \beta(\mathcal{M})$ is a constant integer. We term $\mathcal{M}' = (\bigcup_{B \in \mathcal{B}} B) \cup (\bigcup_{T \in \mathcal{T}} T)$ as the **topological representative** of $\mathcal{M}$.*

Given the classification theorem of 1-manifolds, compact 1-manifolds without boundary are homeomorphic to a circle, while compact 1-manifolds with boundary are homeomorphic to a closed interval. Therefore, the thickened 1-manifolds are homeomorphic to either a solid torus or a ball. Our preliminary analysis focuses on the network size associated with $d$-balls and $d$-tori. Using this as a foundation, we then explore how various topological configurations of thickened 1-manifold impact the size of neural networks. Proposition 1 determines the network size required to approximate a $\mathbb{R}^d$ ball. While the original result is found in Safran & Shamir (2016), our study utilizes fewer parameters and offers a different way to approximate the threshold function. Proposition 2 outlines a network size bound for the approximation of a solid torus.

**Proposition 1** (Approximating a $\mathbb{R}^d$ Ball, adapted from Theorem 2 in Safran & Shamir (2016)). *Given $\epsilon > 0$, there exists a ReLU network $h : \mathbb{R}^d \to \mathbb{R}$ with 3 layers and with size at most $4d^2r^2/\epsilon + 2d + 2$, which can approximate the indicator function $I_{B_r^d}$ within error $R(h) \leq \epsilon$ for any continuous distribution $\mu(\mathbf{x})$.*

**Proposition 2** (Approximating a Solid Torus). *Given $\epsilon > 0$, there exists a ReLU network $h : \mathbb{R}^d \to \mathbb{R}$ with 5 layers and with size at most $\frac{2d}{\epsilon}(4(d-1)(R+r)^2 + 8r^2 + \frac{r}{\sqrt{R-r}}) + 9$, which can approximate the indicator function $I_{T_{r,R}^d}$ within error $R(h) \leq \epsilon$ for any continuous distribution $\mu(\mathbf{x})$.*

Both proofs begin by expressing the indicator function as a composition of truncated power functions. Each truncated power function can be approximated by a piecewise linear function, which can be precisely represented by a ReLU layer. The composition of these power functions is then thresholded, and this threshold function can also be approximated by another ReLU layer. The overall approximation error is estimated across these two steps. Proposition 1 and 2 address the network size associated with approximating $B_r^d$ and $T_{r,R}^d$. Detailed proofs can be found in the Appendix A.3. Building on this, we can infer the network size for approximating topological representatives by combining the complexities of approximating $B_r^d$ and $T_{r,R}^d$ with those of union operation. This consolidated insight is captured in Theorem 1.

**Theorem 1** (Complexity Arising from Topology). *Suppose $\mathcal{M}'$ is the topological representative of $d$-manifold from the thickened 1-manifold family. Given $\epsilon > 0$, there exists a ReLU network $h : \mathbb{R}^d \to \mathbb{R}$ with depth at most $O(\log \beta)$ and size at most $O(\frac{d^2\beta^2}{\epsilon})$, that can approximate the indicator function $I_{\mathcal{M}'}$ with error $R(h) \leq \epsilon$ for any continuous distribution $\mu$ over $\mathbb{R}^d$. $\beta$ is the topological complexity of $\mathcal{M}'$.*

Since a topological representative is the union of balls and solid tori, the overall network size is computed by summing the network sizes required to approximate the balls and tori, along with the network size needed for the union operations. The detailed proof can be found in Appendix A.4. This theorem offers an upper bound on the network size required to approximate the indicator function of a topological representative. It is important to note that this captures the full range of complexities arising from the topology of a manifold $\mathcal{M} \in \mathbb{M}$, given that $\mathcal{M}$ and $\mathcal{M}'$ are homeomorphic. To the best of our knowledge, this is the first result bounding neural network size in terms of a manifold's Betti numbers.

## 4.2 Overall Complexity

For general manifolds in $\mathcal{M} \in \mathbb{M}$, due to their inherent complexity, often defy explicit expression. This hinders the direct use of function analysis for approximating their indicator functions, as was done in previous studies. To tackle this issue, we construct a homeomorphism, a continuous two-way transformation, between $\mathcal{M}$ and its corresponding topological representative $\mathcal{M}'$. This method only alters the geometric properties, preserving the object's topological attributes. Therefore, the network size in approximating the indicator function of $\mathcal{M}$ is constructed by the approximation of the homeomorphism and the classification of topological representatives. The network size of constructing the homeomorphism is exclusively influenced by the geometric properties, whereas the network size of classifying topological representatives pertains solely to topological properties. The latter we already figured in previous section. This methodology enables us to distinguish the influence of topology and geometry of manifold on classifiers. In this section, we aim to obtain the overall network size for a classifier.

To build a homeomorphism from $\mathcal{M}$, we first need to recover the homology of $\mathcal{M}$. The subsequent proposition outlines a lower limit for the number of points essential to recover the homology of the initial manifold $\mathcal{M}$.

**Proposition 3** (Theorem 3.1 in Niyogi et al. (2008))**.** *Let $\mathcal{M}$ be a compact $d$-dimensional submanifold of $\mathbb{R}^D$ with condition number $1/\tau$. Let $X = \{\mathbf{x}_1, \mathbf{x}_2, ..\mathbf{x}_n\}$ be a set of $n$ points drawn in i.i.d. fashion according to the uniform probability measure on $\mathcal{M}$. Let $0 < \epsilon < \frac{\tau}{2}$. Let $U = \bigcup_{\mathbf{x} \in X} B_\epsilon(\mathbf{x})$ be a corresponding random open subset of $\mathbb{R}^D$. Then for all*

$$n > \lambda_1 (log(\lambda_2) + log(\frac{1}{\delta})), \tag{8}$$

*$U$ is a $\epsilon$-cover of $\mathcal{M}$, and the homology of $U$ equals the homology of $\mathcal{M}$ with high confidence (probability $> 1 - \delta$). Here*

$$\lambda_1 = \frac{vol(\mathcal{M})}{(cos^d\theta_1)vol(B^d_{\epsilon/4})} \;\; and \;\; \lambda_2 = \frac{vol(\mathcal{M})}{(cos^d\theta_2)vol(B^d_{\epsilon/8})}, \tag{9}$$

*$\theta_1 = arcsin(\epsilon/8\tau)$ and $\theta_2 = arcsin(\epsilon/16\tau)$. $vol(B^d_\epsilon)$ denotes the $d$-dimensional volume of the standard $d$-dimensional ball of radius $\epsilon$. $vol(\mathcal{M})$ is the $d$-dimensional volume of $\mathcal{M}$.*

This result stipulates a lower bound for the training set size necessary to recover the homology of the manifold, which is the foundation to learn the homeomorphism between a manifold $\mathcal{M} \in \mathbb{M}$ and its topological representative $\mathcal{M}'$. However, directly constructing this homeomorphism remains challenging. As a workaround, we develop a simplicial homeomorphism to approximate the genuine homeomorphism. Notably, this simplicial approach lends itself readily to representation via neural networks.

Combined with the topological representative classification network in Theorem 1, we can construct a classification network for general manifolds in $\mathbb{M}$, as depicted in Figure 4. Initially, we project $\mathcal{M}$ to its simplicial approximation $|K|$ using a neural network $N_p$. This is succeeded by a network $N_\phi$ that facilitates the simplicial homeomorphism between $|K|$ and $|L|$, the latter being the simplicial approximation of the topological representative $\mathcal{M}'$. Finally, a network $h$ is utilized to classify between $|L_1|$ and $|L_0|$. Consequently, the network's size is divided into two main parts: one focused on complexities related to geometric attributes and the other concerning topological aspects. This distinction separates topology from geometry in classification problems.

In Theorem 2, we design such a neural network based on this training set, ensuring that approximation errors are effectively controlled. The detailed proof is provided in Appendix A.5. Our proof strategy begins with

the construction of a ReLU network, followed by an evaluation of the network's size. Subsequently, we place bounds on the involved approximation errors.

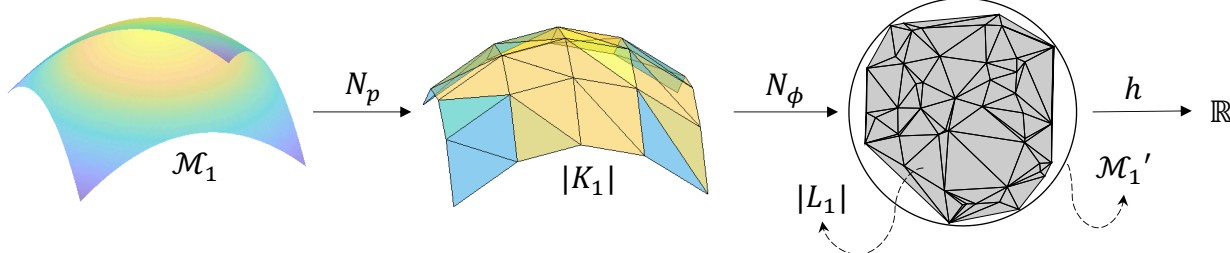

Figure 4: Construction of the network $g$. The network first learns a low-dimensional embedding and then performs classification in the embedding space. This paradigm mirrors the typical operation of deep networks. While the diagram illustrates only the process for the manifold of the positive class, the procedure for $\mathcal{M}_0$ mirrors this operation identically.

**Theorem 2** (Main Theorem). *Let $\mathcal{M} = \mathcal{M}_1 \sqcup \mathcal{M}_0 \subset \mathbb{R}^D$ be a $d$-dimensional manifold from the thickened 1-manifold family. $\mathcal{M}_1$ and $\mathcal{M}_0$ are two disjoint sub-manifolds of $\mathcal{M}$ representing two classes. The condition number of $\mathcal{M}$ is $\frac{1}{\tau}$ and the total Betti number of $\mathcal{M}_1$ is $\beta$. Given a training set $\{(\mathbf{x}_i, y_i) | \mathbf{x}_i \in \mathcal{M}, y_i \in \{0,1\}\}_{i=1}^n$, where $\mathbf{x}_i$ are sampled i.i.d. from $\mathcal{M}$ by a uniform distribution, and $y_i = I_{\mathcal{M}_1}(\mathbf{x}_i)$. For any $\delta > 0$, if inequality (8) holds, then for any $\epsilon > 0$, there exists a ReLU network $g$ with depth at most $O(\log \beta + d \log \frac{1}{\tau} + \log \log \frac{1}{\tau \delta})$ and size at most $O(\frac{d^2 \beta^2}{\epsilon} + \tau^{-d^2/2} \log^{d/2} \frac{1}{\tau \delta} + D\tau^{-d} \log \frac{1}{\tau \delta})$, such that*

$$P(R(g) \le \epsilon) > 1 - \delta, \tag{10}$$

*where $R(g) = \int_{\mathcal{M}} (g - I_{\mathcal{M}_1})^2 \mu(\mathbf{x}) d\mathbf{x}$ with any continuous distribution $\mu$.*

*Proof Sketch.* Since $\mathcal{M} = \mathcal{M}_1 \sqcup \mathcal{M}_0$ is from thickened 1-manifold family, it has a topological representative $\mathcal{M}' = \mathcal{M}'_1 \sqcup \mathcal{M}'_0 \subset \mathbb{R}^d$, where $\mathcal{M}'_1$ and $\mathcal{M}'_0$ are topological representatives of $\mathcal{M}_1$ and $\mathcal{M}_0$, respectively. The proof follows Figure 4, by first constructing simplicial approximations $|K|$ and $|L|$ of $\mathcal{M}$ and $\mathcal{M}'$, respectively. Then we represent a simplicial homeomorphism $\phi : |K| \to |L|$ by a neural network $N_\phi$, where $K$ is constructed from $\mathcal{M}$ and $L$ from $\mathcal{M}'$. Built on the top of this, a projection from $\mathcal{M}$ to its simplicial approximation $|K|$ is represented by another network $N_p$. The overall network can be constructed by $g = h \circ N_\phi \circ N_p$. The proof is completed by first constructing the network $g$, and then bounding the approximation error.

Upon examining the depth and size of the neural network, it becomes evident that the topological complexity, denoted by $\beta$, and the geometric complexity, symbolized by $\tau$, are distinctly delineated. Note that our result provides an upper bound on the network size required to achieve a given classification error rate and may not necessarily be tight. However, our empirical validation suggests that the topological component of the bound has the potential to be tight in practice. While this bound serves as a theoretical guarantee, it also offers valuable insights into the distinct factors that contribute to the complexity of the network. The topological complexity contributes $O\left(\frac{d^2 \beta^2}{\epsilon}\right)$ to the overall network size. In contrast, geometry contributes $O\left(\tau^{-d^2/2} \log^{d/2} \frac{1}{\tau \delta} + D\tau^{-d} \log \frac{1}{\tau \delta}\right)$.

## 5 Experiments

In this section, we present numerical results that showcase our topological bound $O\left(\frac{d^2 \beta^2}{\epsilon}\right)$ in fixed dimension. Even though this bound is derived through construction and serves as an upper limit, it is intriguing to discover that the bound is tight and can be readily observed in experimental settings. In practical scenarios, when training networks of varying sizes on data drawn from manifolds with different total Betti numbers $\beta$, and achieving the same error rate, we anticipate a linear relationship between the network size and $\beta^2$.

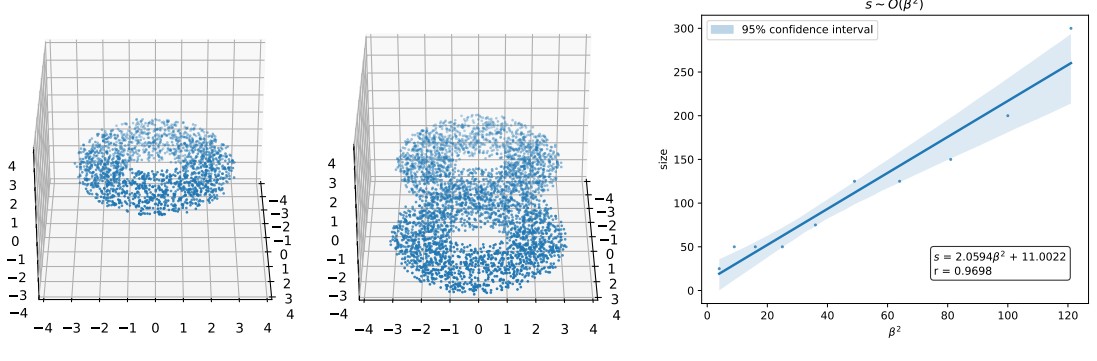

(a) Solid torus of genus 1.     (b) Solid torus of genus 2.     (c) Relation between network size and $\beta^2$.

Figure 5: Validation of topological complexity bound. We utilize manifolds characterized as solid genus-$g$ tori, with $g$ spanning from 1 to 10. For each torus, we consistently sample $g \times 10^4$ points from a surrounding bounding box that cover the torus. The labels for these points are generated using the indicator function of the solid torus. (a,b) are two examples of genus-$g$ tori, but *samples from the background class are not visualized in the graph*. (c) showcases the linear regression results between the network size and the square of topological complexity. A 5-layer neural network with adaptive width is trained to fit tori of varying genus $g$. The width of the network is increased until the training accuracy can exceeds 0.95 when the MSE loss converges. This regression underscores a pronounced linear association between network size $s$ and $\beta^2$, with a correlation coefficient 0.9698.

We utilize manifolds characterized as solid genus-$g$ tori, with $g$ spanning from 1 to 10. Each genus-$g$ torus is synthesized by overlapping two identical tori. For each torus, we consistently sample $g \times 10^4$ points from a surrounding bounding box. The labels for these points are generated using the indicator function of the solid torus.

For training, we deploy a 5-layer ReLU network, gradually increasing its width until the training accuracy surpasses 0.95. Figure 5c presents a regression line charting the relationship between network size and the squared topological complexity, $\beta^2$. This regression underscores a pronounced linear association between network size $s$ and $\beta^2$, with a correlation coefficient 0.9698.

The tori used in the previous experiment were sampled using Equation 7 with parameters $R = 3$ and $r = 1$. To investigate the impact of geometric properties, we shrink the inner radius to $r = 0.5$, effectively increasing the condition number $\frac{1}{\tau}$. We then repeat the same experiments on these modified tori for genus 1 to 10, as shown in Figure 6. The results continue to indicate a linear relationship between network size and $\beta^2$. Compared to Figure 5c, learning a torus with the same genus but a larger condition number requires a larger network, highlighting the impact of geometric complexity on network expressivity.

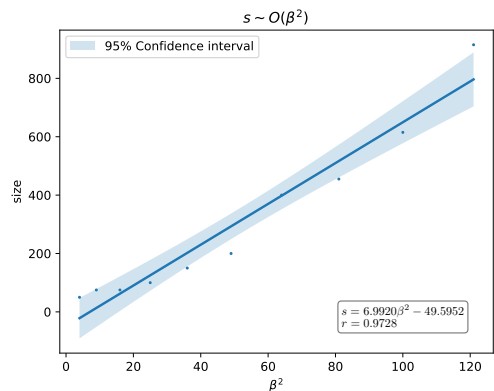

Figure 6: Relation between network size and $\beta^2$ with larger condition number.

## 6  Conclusion

In this study, we delved into the intricate relationship between network size, and both geometric and topological characteristics of manifolds. Our findings underscored that while many existing studies have been focused on geometric intricacies, it is important to also appreciate the manifold's topological characteristics. These characteristics not only offer an alternative perspective on data structures but also influence network size in significant ways.

Our proposed network size bounds represent theoretical upper limits, meaning that real-world implementations may yield efficiencies beyond these confines. To attain a more direct and refined theoretical bound, we may need more comprehensive descriptors of manifolds that go beyond merely the Betti numbers and the condition number. We leave this exploration for future work. We hope that our study acts as a catalyst for further research, pushing the boundaries of manifold learning and its applications in modern AI systems.

**Acknowledgments**

This research is partially supported by National Science Foundation (NSF) grants CCF-2144901, the National Institue of Health (NIH) grants R01NS 143143, and the Stony Brook Trustees Faculty Award. This research is also partially supported by NSF CCF-2503086.

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

# A   Appendix

In this section, we formally prove the theoretical findings presented in the primary manuscript. Initially, we utilize some necessary definitions and existing results. Then we prove the network size bound for balls, solid tori and general topological representatives.

## A.1   Additional Definitions

The first definition is *Betti number*, which is a vital part of this paper. The $k$-th Betti number is defined as the rank of $k$-th homology group. Therefore, we have to properly define homology group first. Our definition follows Hatcher (2002) but is tailed for simplification. We first define *simplicial homology* for simplicial complexes. (Actually for $\Delta$-complexes. we simplify it to avoid introducing $\Delta$-complexes.) Then extend it to *singular homology* that can be applied to manifolds.

**Simplicial Homology.** Let $K$ be a simplicial complex, and let $K^k$ be the set of all $k$-dimensional simplices in $K$. The set of $K^k$ together with the field $\mathbb{Z}_2$ forms a group $C_k(K)$. It is a vector space defined on $\mathbb{Z}_2$ with $K^k$ as a basis. The element of $C_k(K)$ is called a $k$-chain. Let $\sigma \in K^k$ be a $k$-simplex. The boundary $\partial_k(\sigma)$ is the collection of its $(k-1)$-dimensional faces, which is a $k-1$-simplicial complex. The boudnary operator is linear, i.e.

$$\partial_k(z_1\sigma_1 + z_2\sigma_2) = z_1\partial_k(\sigma_1) + z_2\partial_k(\sigma_2).$$

The boundary operator $\partial_k : C_k(K) \to C_{k-1}(K)$ introduces a chain complex

$$\cdots \to C_d \xrightarrow{\partial_d} C_{d-1} \xrightarrow{\partial_{d-1}} C_{d-2} \to \cdots \to C_0 \xrightarrow{\partial_0} \emptyset.$$

$d$ is the maximum dimension of $K$. $\operatorname{Ker}\partial_k$ is the collection of $k$-chains with empty boundary and $\operatorname{Im}\partial_k$ is the collection of $(k-1)$-chains that are boundaries of $k$-chains. Then we can define the $k$-th homology group of the chain complex to be the quotient group $H_k = \operatorname{Ker}\partial_k/\operatorname{Im}\partial_{k+1}$. The $k$-th Betti number is defined by

$$\beta_k = \operatorname{rank} H_k.$$

**Singular Homology.** Given a topological space $X$, the $k$-th singular chain group $C_k(X)$ is defined as the free Abelian group generated by the continuous maps $\phi : K^k \to X$, where $K^k$ is the standard $k$-simplex in $\mathbb{R}^k$. Each such map is referred to as a singular $k$-simplex in $X$.

A boundary operator $\partial_k : C_k(X) \to C_{k-1}(X)$ can be defined as:

$$\partial\phi = \sum_{i=0}^{n}(-1)^i\phi|_{[v_0,\cdots,\hat{v}_i,\cdots,v_n]},$$

where $\phi|_{[v_0,\cdots,\hat{v}_i,\cdots,v_n]}$ represents the restriction of $\sigma$ to the $i$-th face of $K^k$.

The $k$-th singular homology group $H_k(X)$ is then represented as the quotient:

$$H_k(X) = \operatorname{Ker}\partial_k/\operatorname{Im}\partial_{k+1}.$$

The $k$-th Betti number is still defined as $\beta_k = \operatorname{rank} H_k(X)$.

**Connected Sum of Thickened 1-Manifold.** Let $\mathcal{M} = \mathcal{M}^1 \times B^d$ and $\mathcal{N} = \mathcal{N}^1 \times B^d$ be two $d$-thickened 1-manifolds, where $\mathcal{M}^1$ and $\mathcal{N}^1$ are compact 1-manifolds (with or without boundary), and $B^d$ is a $d$-dimensional closed ball. We define the *connected sum* $\mathcal{M}\#\mathcal{N}$ by performing connected sum on their boundaries and then extend the gluing to the interior. The process is defined as:

1. The connected sum of boundaries. The boundaries of $\mathcal{M}$ and $\mathcal{N}$ are closed manifolds, where $\partial\mathcal{M} = \mathcal{M}^1 \times S^{d-1}$ and $\partial\mathcal{N} = \mathcal{N}^1 \times S^{d-1}$. Two open neighborhoods $U \subset \partial\mathcal{M}$ and $V \subset \partial\mathcal{N}$ are removed from the boundaries. Then a homeomorphism $h : \partial U \to \partial V$ is defined to match the resulting boundaries. Then glue $\partial\mathcal{M} \setminus U$ and $\partial\mathcal{N} \setminus V$ along $\partial U$ and $\partial V$ via $h$, forming $\partial(\mathcal{M}\#\mathcal{N})$.

2. Extend the gluing to the interiors of $\mathcal{M}$ and $\mathcal{N}$ by identifying corresponding points in the neighborhoods adjacent to $U$ and $V$.

## A.2 Preliminary Results

We present some pre-established results regarding the network size associated with learning a 1-dimensional piecewise linear function, as well as basic combinations of functions.

**Lemma 2** (Theorem 2.2. in Arora et al. (2018)). *Given any piecewise linear function $\mathbb{R} \to \mathbb{R}$ with $p$ pieces there exists a 2-layer ReLU network with at most $p$ nodes that can represent $f$. Moreover, if the rightmost or leftmost piece of a the piecewise linear function has $0$ slope, then we can compute such a $p$ piece function using a 2-layer ReLU network with size $p - 1$.*

**Lemma 3** (Function Composition, Lemma D.1. in Arora et al. (2018)). *If $f_1 : \mathbb{R}^d \to \mathbb{R}^m$ is represented by a ReLU DNN with depth $k_1 + 1$ and size $s_1$, and $f_2 : \mathbb{R}^m \to \mathbb{R}^n$ is represented by a ReLU DNN with depth $k_2 + 1$ and size $s_2$, then $f_2 \circ f_1$ can be represented by a ReLU DNN with depth $k_1 + k_2 + 1$ and size $s_1 + s_2$.*

**Lemma 4** (Function Addition, Lemma D.2. in Arora et al. (2018)). *If $f_1 : \mathbb{R}^n \to \mathbb{R}^m$ is represented by a ReLU DNN with depth $k + 1$ and size $s_1$, and $f_2 : \mathbb{R}^n \to \mathbb{R}^m$ is represented by a ReLU DNN with depth $k + 1$ and size $s_2$, then $f_1 + f_2$ can be represented by a ReLU DNN with depth $k + 1$ and size $s_1 + s_2$.*

**Lemma 5** (Taking maximums, Lemma D.3. in Arora et al. (2018)). *Let $f_1, ..., f_m : \mathbb{R}^n \to \mathbb{R}$ be the functions that each can be represented by ReLU networks with depth $k_i + 1$ and size $s_i$, $i = 1, ..., m$. Then the function $f : \mathbb{R}^n \to \mathbb{R}$ defined as $f = \max\{f_1, ..., f_m\}$ can be represented by a ReLU network of depth at most $\max\{k1, ..., k_m\} + \log(m) + 1$ and size at most $s_1 + ... + s_m + 4(2m - 1)$.*

We proceed to disclose the network size involved in learning a 1-dimensional Lipschitz function.

**Lemma 6** (Lipschitz Function Approximation, adapted from Lemma 11 in Eldan & Shamir (2016)). *For any $L$-Lipschitz function $f : \mathbb{R} \to \mathbb{R}$ which is constant outside a bounded interval $[a, b]$, and for any $\epsilon > 0$, there exits a two-layer ReLU network $h(x)$ with at most $\lceil L(b - a)/\epsilon \rceil + 1$ nodes, such that*

$$\sup_{x \in \mathbb{R}} |f(x) - h(x)| < \epsilon.$$

*Proof.* We follow the original proving idea but adapt it for better understanding. We prove the lemma by estimate the Lipschitz function by a piece-wise linear function within error $\epsilon$ and use a two-layer ReLU network to represent the piece-wise linear function.

We first cut the interval equally into $m$ sections $[a, b] = \bigcup_{i=1}^{m}[a + (i-1)\delta, a + i\delta]$, where $\delta = (b - a)/m$. For each interval $I_i = [a + (i-1)\delta, a + i\delta]$, we denote $f_i(x) = f|_{I_i}$. Then $\forall x_1, x_2 \in I_i$, $|f_i(x_1) - f_i(x_2)| \leq L|x_1 - x_2| \leq L\delta$. Let $h_i(x)$ be the linear function defined on this interval and connect $(a + (i-1)\delta, f_i(a + (i-1)\delta))$ and $(a + i\delta, f_i(a + i\delta))$. Then we can bound the difference between $f_i(x)$ and $h_i(x)$ by

$$\begin{aligned} |f_i(x) - h_i(x)| &\leq \max\{|\max f_i(x) - \min h_i(x)|, |\min f_i(x) - \max h_i(x)|\} \\ &= \max\{|\max f_i(x) - f_i(a + (i-1)\delta)|, |\min f_i(x) - f_i(a + i\delta))|\} \\ &\leq L\delta. \end{aligned} \tag{A.1}$$

The second line assumes $h_i(x)$ is non-decreasing. The other case can also be easily verified. By setting $m = \lceil \frac{L(b-a)}{\epsilon} \rceil$, for every interval, the error is controlled by $\epsilon$. Let $h(x)$ be the collection of all $h_i$ and also the constant outside of $[a, b]$, so we have $\sup_{x \in \mathbb{R}} |f(x) - h(x)| < \epsilon$.

$h(x)$ is a piece-wise linear function with $m + 2$ pieces. According to Lemma 2, there exists a 2-layer ReLU network with at most $m + 1$ pieces that can represent $h(x)$. Proof done. $\square$

## A.3 Approximating Basic Solid Manifolds

Now we are in a good position to prove Proposition 1 and 2.

**Proposition 1** (Approximating a $\mathbb{R}^d$ Ball, adapted from Theorem 2 in Safran & Shamir (2016)). *Given $\epsilon > 0$, there exists a ReLU network $h : \mathbb{R}^d \to \mathbb{R}$ with 3 layers and with size at most $4d^2r^2/\epsilon + 2d + 2$, which can approximate the indicator function $I_{B_r^d}$ within error $R(h) \leq \epsilon$ for any continuous distribution $\mu(\mathbf{x})$.*

*Proof.* We generally follow the original proof but derive a slightly different bound with fewer parameters. The proof is organized by first using a non-linear layer to approximate a truncated square function and then using another non-linear layer to approximate a threshold function. Consider the truncated square function

$$l(x; r) = \min\{x^2, r^2\}. \tag{A.2}$$

Clearly $l(x; r)$ is a Lipschitz function with Lipschitz constant $2r$. Applying Lemma 6, we have a 2-layer ReLU network $h_{11}$ that can approximate $l(x; r)$ with

$$\sup_{x \in \mathbb{R}} \left| h_{11}(x) - l(x) \right| \le \epsilon_1, \tag{A.3}$$

with at most $2r^2/\epsilon_1 + 2$ nodes. Now for $\mathbf{x} \in \mathbb{R}^d$, let

$$h_1(\mathbf{x}) = \sum_{i=1}^{d} h_{1i}(x_i). \tag{A.4}$$

Note that $h_1$ is also a 2-layer network because no extra non-linear operation is introduced in equation A.4, and has size at most $2dr^2/\epsilon_1 + 2d$. This can also be verified by Lemma 4. Let

$$L(\mathbf{x}) = \sum_{i}^{d} L(x_i; r), \tag{A.5}$$

and we have

$$\sup_{\mathbf{x}} \left| h_1(\mathbf{x}) - L(\mathbf{x}) \right| \le d\epsilon_1. \tag{A.6}$$

Let $\epsilon_1 = d\epsilon_1$, then $h_1$ has size at most $2d^2r^2/\epsilon_1 + 2d$. Although $L(\mathbf{x})$ is different from $\sum x_i^2$, the trick here is to show $B_r^d = \{\mathbf{x} : L(\mathbf{x}) \le r^2\}$.

On the one hand, if $L(rvx) \le r^2$, remember that

$$L(\mathbf{x}) = \sum_{i=1}^{d} \min\{x_1^2, r^2\} \le r^2. \tag{A.7}$$

This means for all $x_i$, $x_i \le r^2$. Therefore, $L(\mathbf{x}) = \sum_{i=1}^{d} x_i^2$. On the other, $L(\mathbf{x}) > r^2$ only happens when there exists a $i$, such that $x_i^2 > r^2$. Thus, $\mathbf{x} \notin B_r^d$. Consequently, one can represent $I_{B_r^d}$ by $L(\mathbf{x}) \le r^2$.

The next step towards this proposition is to construct another 2-layer ReLU network to threshold $L(\mathbf{x})$. Consider

$$f(x) = \begin{cases} 1, & x < r^2 - \delta, \\ \frac{r^2 - x}{\delta}, & x \in [r^2 - \delta, r^2], \\ 0, & x > r^2. \end{cases} \tag{A.8}$$

Note that $f$ is a 3-piece piece-wise linear function that approximates a threshold function. According to Lemma 2, a 2-layer ReLU network $h_2$ with size 2 can represent $f$. The function $f \circ L(\mathbf{x})$ can then be estimated by a 3-layer network $h = h_2 \circ h_1$, whose size is $2d^2r^2/\epsilon_1 + 2d + 2$. The next step is to bound the error between $h$ and $I_{B_r^d}$. We consider the $L_2$-type bound $||h(\mathbf{x}) - I_{B_r^d}(\mathbf{x})||_{L_2(\mu)} = \int_{\mathbb{R}^d} (h(\mathbf{x}) - I_{B_r^d}(\mathbf{x}))^2 \mu(\mathbf{x}) d\mathbf{x}$. We divide the integral into two parts

$$\begin{aligned} &||h(\mathbf{x}) - I_{B_r^d}(\mathbf{x})||_{L_2(\mu)} \\ &\le ||f \circ L(\mathbf{x}) - I_{B_r^d}(\mathbf{x})||_{L_2(\mu)} + ||f \circ L(\mathbf{x}) - h_2 \circ h_1(\mathbf{x})||_{L_2(\mu)} \\ &= I_1 + I_2. \end{aligned} \tag{A.9}$$

Since $\mu(\mathbf{x})$ is continuous, there exists $\delta$ such that

$$\int_{S_\delta} \mu(\mathbf{x}) d\mathbf{x} \le \epsilon_2. \tag{A.10}$$

$S_\delta = \{\mathbf{x} \in \mathbb{R}^d : r^2 - \delta \leq \sum_{i=1}^d x_i^2 \leq r^2\}$. Combine equation A.8 we have

$$
\begin{aligned}
I_1 &= \int_{\mathbb{R}^d} (f \circ L(\mathbf{x}) - I_{B_r^d}(\mathbf{x}))^2 \mu(\mathbf{x}) d\mathbf{x} \\
&= \int_{S_\delta} (f \circ L(\mathbf{x}) - I_{B_r^d}(\mathbf{x}))^2 \mu(\mathbf{x}) d\mathbf{x} \\
&= \int_{S_\delta} (f \circ L(\mathbf{x}) - 1)^2 \mu(\mathbf{x}) d\mathbf{x} \\
&\leq \int_{S_\delta} \mu(\mathbf{x}) d\mathbf{x} \\
&\leq \epsilon_2.
\end{aligned}
\tag{A.11}
$$

The first inequality is because $f \in [0,1]$, such that $(f \circ L(\mathbf{x}) - 1)^2 \leq 1$. The second part of the error can be easily bounded by its infinity norm.

$$
I_2 = ||f \circ L(\mathbf{x}) - h_2 \circ h_1(\mathbf{x})||_{L_2(\mu)} \leq ||f \circ L(\mathbf{x}) - h_2 \circ h_1(\mathbf{x})||_\infty \leq \epsilon_1.
\tag{A.12}
$$

The last inequality is because $h_2$ is the exact representation of $f$, the error only occurs between $L(\mathbf{x})$ and $h_1$. Combine A.11 and A.12, and let $\epsilon_1 = \epsilon_2 = \epsilon/2$, we have

$$
||h(\mathbf{x}) - I(\mathbf{x})||_{L_2(\mu)} \leq \epsilon.
\tag{A.13}
$$

The size of network $h$ is then bounded by $4d^2 r^2/\epsilon + 2d + 2$. $\qquad\square$

**Proposition 2** (Approximating a Solid Torus). *Given $\epsilon > 0$, there exists a ReLU network $h : \mathbb{R}^d \to \mathbb{R}$ with 5 layers and with size at most $\frac{2d}{\epsilon}(4(d-1)(R+r)^2 + 8r^2 + \frac{r}{\sqrt{R-r}}) + 9$, which can approximate the indicator function $I_{T_{r,R}^d}$ within error $R(h) \leq \epsilon$ for any continuous distribution $\mu(\mathbf{x})$.*

*Proof.* The proof is done by two steps. We first use layers to estimate a truncated function. Then estimate a threshold function by another layer.

Consider the truncated square function and root function,

$$
\begin{aligned}
l_1(x; \gamma) &= \min\{x^2, \gamma^2\}, \\
l_2(x; \gamma_1, \gamma_2) &= \min\{\max\{\sqrt{x}, \gamma_1\}, \gamma_2\}, (\gamma_1 < \gamma_2).
\end{aligned}
$$

The Lipschitz constants for $l_1$ and $l_2$ are $2\gamma$ and $\frac{1}{2\sqrt{\gamma_1}}$, respectively. By Lemma 6, there is a 2-layer ReLU network to approximate $l_1$ and $l_2$ with size $\lceil 4\gamma^2/\epsilon_1 \rceil + 1$ and $\lceil (\gamma_2 - \gamma_1)/(2\epsilon_1 \sqrt{\gamma_1}) \rceil + 1$, respectively. Let

$$
L(\mathbf{x}) = l_1(x_d; r) + l_1(l_2(\sum_{i=1}^{d-1} l_1(x_i; R+r); R-r, R+r) - R; r).
\tag{A.14}
$$

Then it is time to show $T_{r,R}^d = \left\{ \mathbf{x} \in \mathbb{R}^d : x_d^2 + \left( \sqrt{\sum_{i=1}^{d-1} x_i^2} - R \right)^2 \leq r^2 \right\} = \{\mathbf{x} : L(\mathbf{x}) \leq r^2\}$. For $\mathbf{x} \in I_{T_{r,R}^d}(\mathbf{x})$, the following inequalities hold

$$
x_i^2 \leq (R+r)^2, 1 \leq i \leq d-1,
\tag{A.15}
$$

$$
R + r \geq \sqrt{\sum_{i=1}^{d-1} x_i^2} \geq R - r,
\tag{A.16}
$$

$$
x_d^2 \leq r^2, (\sqrt{\sum_{i=1}^{d-1} x_i^2} - R)^2 \leq r^2.
\tag{A.17}
$$

These indicate that $L(\mathbf{x}) = x_d^2 + \left(\sqrt{\sum_{i=1}^{d-1} x_i^2} - R\right)^2 \le r^2$, when $\mathbf{x} \in T_{r,R}^d$. And when $\mathbf{x} \notin T_{r,R}^d$, if $L(\mathbf{x}) = x_d^2 + \left(\sqrt{\sum_{i=1}^{d-1} x_i^2} - R\right)^2$ still holds, clearly $L(\mathbf{x}) > r^2$. Otherwise, one of the inequalities in A.15, A.16 and A.17 must break. If one of A.17 breaks, then clearly $L(\mathbf{x}) > r^2$. If A.16 does not hold, then $(\sqrt{x_1^2 + x_2^2} - R)^2 > r^2$, resulting $L(\mathbf{x}) > r^2$. The violation of A.15 resulting violation of A.16, which then leads to $L(\mathbf{x}) > r^2$.

To see how a ReLU network can estimate $L(\mathbf{x})$, we start by estimating each of its component. We define the following 2-layer networks. To make the overall network take $\mathbf{x} \in \mathbb{R}^d$ as input, we consider the structure in figure A.1.

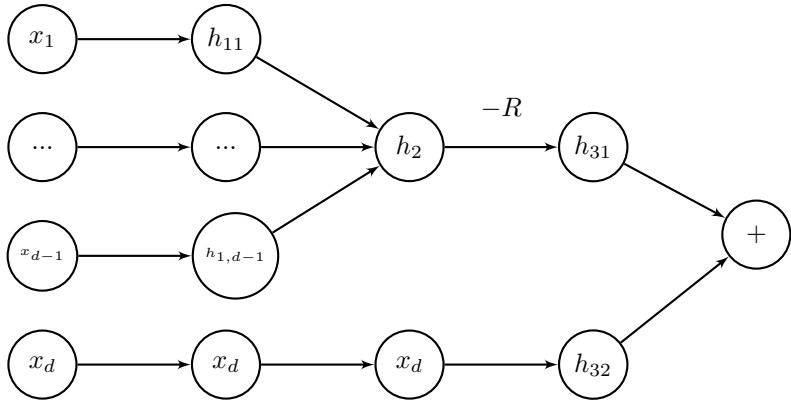

Figure A.1: Network construction.

The size of parts in the network is provided in table A.1.

Table A.1: Sub-network Size.

| Network | Target | Size |
|---|---|---|
| $h_{1i}$ | $l_1(x_i; R+r)$ | $s_{1i} = \lceil 4(R+r)^2/\epsilon_1 \rceil + 1$ |
| $h_2$ | $l_2(x; R-r, R+r)$ | $s_2 = \lceil r/(\epsilon_1 \sqrt{R-r}) \rceil + 1$ |
| $h_{31}$ | $l_1(x; , r)$ | $s_{31} = \lceil 4r^2/\epsilon_1 \rceil + 1$ |
| $h_{32}$ | $l_1(x_d; , r)$ | $s_{32} = \lceil 4r^2/\epsilon_1 \rceil + 1$ |

By Lemma 4 and Lemma 3 and the given structure, a ReLU network $\tilde{L}$ with depth 4 and size $(d-1)s_{11} + s_2 + s_{31} + s_{32} + 2$, where $d = 3$, can approximate $L(\mathbf{x})$ such that

$$\sup_{\mathbf{x}} |L(\mathbf{x}) - \tilde{L}(\mathbf{x})| \le d\epsilon_1. \tag{A.18}$$

The next step is to threshold $L(\mathbf{x})$. Consider a function

$$f(x) = \begin{cases} 1, & x < r^2 - \delta, \\ \frac{r^2 - x}{\delta}, & x \in [r^2 - \delta, r^2], \\ 0, & x > r^2. \end{cases} \tag{A.19}$$

This function approximates a thresholding function $I[x \le r^2]$ but with error inside the interval $[r^2 - \delta, r^2]$. By Lemma 2, a 2-layer ReLU network $\tilde{f}$ with size 2 can represent $f(x)$. Then $\tilde{f} \circ \tilde{L}$ is a ReLU network with depth 5 and size $(d-1)s_{11} + s_2 + s_{31} + s_{32} + 4$, such that

$$\sup_{\mathbf{x}} |f \circ L(\mathbf{x}) - \tilde{f} \circ \tilde{L}(\mathbf{x})| \le \epsilon_1, \tag{A.20}$$

with letting $\epsilon_1 = \epsilon_1/d$.

Let $h(\mathbf{x}) = \tilde{f} \circ \tilde{L}(\mathbf{x})$. We claim that $h(\mathbf{x})$ is the desired network with depth 5 and size

$$
\begin{aligned}
&(d-1)s_{11} + s_2 + s_{31} + s_{32} + 4 \\
&= \frac{d}{\epsilon_1}\left(4(d-1)(R+r)^2 + 8r^2 + \frac{r}{\sqrt{R-r}}\right) + 9 \\
&= O\left(\frac{d^2}{\epsilon_1}\right).
\end{aligned}
\tag{A.21}
$$

To finalize our proof, we just need to bound the error $||h(\mathbf{x}) - I_{T_{r,R}^d}(\mathbf{x})||_{L_2(\mu)}$. The proof follows proof of Proposition 1. The error is divided into two parts and is bounded separately. The only difference is we define $S_\delta$ to be $S_\delta = \left\{\mathbf{x} \in \mathbb{R}^d : r^2 - \delta \le x_d^2 + \left(\sqrt{\sum_{i=1}^{d-1} x_i^2} - R\right)^2 \le r^2\right\}$, such that

$$
\int_{S_\delta} \mu(\mathbf{x})d\mathbf{x} \le \epsilon_2.
\tag{A.22}
$$

We can get $I_1 \le \epsilon_2$, and $I_2 \le \epsilon_1$. Let $\epsilon_1 = \epsilon_2 = \epsilon/2$, we have

$$
||h(\mathbf{x}) - I(\mathbf{x})||_{L_2(\mu)} \le \epsilon.
\tag{A.23}
$$

And $h$ has size at most $\frac{2d}{\epsilon}\left(4(d-1)(R+r)^2 + 8r^2 + \frac{r}{\sqrt{R-r}}\right) + 9$. $\qquad\square$

## A.4 Approximating Topological Representatives

After getting the size arising from fundamental manifolds, we proceed to study the combination of them. We start by proving the representative property.

**Lemma 1** (Topological Representative). *Let $\mathcal{M} \subset \mathbb{M}$ be a $d$-dimensional manifold from the thickened 1-manifold family. There exist a set of $m_1$ $d$-balls $\mathcal{B} = \{B_{r_i}^d(\mathbf{c}_i)\}_{i=1}^{m_1}$ and a set of $m_2$ solid $d$-tori $\mathcal{T} = \{T_{r_i, R_i}^d(\mathbf{c}_i)\}_{i=1}^{m_2}$, such that $\mathcal{M}$ is homeomorphic to the union $\left(\bigcup_{B \in \mathcal{B}} B\right) \cup \left(\bigcup_{T \in \mathcal{T}} T\right)$, where $m_1 + m_2 \le \beta(\mathcal{M})$ is a constant integer. We term $\mathcal{M}' = \left(\bigcup_{B \in \mathcal{B}} B\right) \cup \left(\bigcup_{T \in \mathcal{T}} T\right)$ as the **topological representative** of $\mathcal{M}$.*

*Proof.* Since $\mathcal{M} \subset \mathbb{M}$, regarding the definition, $\mathcal{M}$ is homeomorphic to connect sum or disjoint union of $m$ thickened 1-manifolds. A thickened 1-manifold is denoted as $\mathcal{M}^1 \times B^{d-1}$. And based on the classification theorem of 1-manifold, $\mathcal{M}^1$ is homeomorphic either to a circle $S^1$ or $I = [0,1]$. If $\mathcal{M}^1 \sim S^1$, then $\mathcal{M}^1 \times B^{d-1} \sim S^1 \times B^{d-1} \sim T_{r,R}^d$. If $\mathcal{M}^1 \sim I$, then $\mathcal{M}^1 \times B^{d-1} \sim I \times B^{d-1} \sim B_r^d$. Now let $\mathcal{M}_1$ and $\mathcal{M}_2$ be two thickened 1-manifold, the connected sum of $\mathcal{M}_1$ and $\mathcal{M}_2$, denoted by $\mathcal{M}_1 \oplus \mathcal{M}_2$, can be represented by the union of two manifolds $\mathcal{M}_1 \cup \mathcal{M}_2$. Since $\mathcal{M}$ is homeomorphic to connect sum or disjoint union of $m$ thickened 1-manifolds, suppose $m_1$ of them are homeomorphic to $B_r^d$ and $m_2$ of them are homeomorphic to $T_{r,R}^d$. Therefore, $\mathcal{M} \sim \left(\bigcup_{B \in \mathcal{B}} B\right) \cup \left(\bigcup_{T \in \mathcal{T}} T\right)$.

Notice that among $\mathcal{B}$, the union is disjoint union, otherwise the union is still homeomorphic to a $B_r^d$. Similarly, the union between $\mathcal{B}$ and $\mathcal{T}$ is also disjoint union, otherwise the union is still homeomorphic to a solid torus $T_{r,R}^d$. The union among $\mathcal{T}$ could be joint union or disjoint union. For two solid tori $T_1^d$ and $T_2^d$, $\beta(T_1^d) = \beta(T_1^d) = 2$. $\beta(T_1^d \cup T_2^d) \ge 3 > 2$. Overall, $\beta(\mathcal{M}) \ge m_1 + m_2$.

$\qquad\square$

**Lemma 7** (Manifold Union). *$\mathcal{M}_1$ and $\mathcal{M}_2$ are two manifolds in $\mathbb{R}^d$. $I_{\mathcal{M}_1}$ can be approximated by a ReLU network $h_1$ with depth $d_1 + 1$ and size at most $s_1$ with error $R(h_1) < \epsilon_1$, $I_{\mathcal{M}_2}$ can be approximated by a ReLU network $h_2$ with depth $d_2 + 1$ and size at most $s_2$ with error $R(h_2) < \epsilon_2$. Then $I_{\mathcal{M}_1 \cup \mathcal{M}_2}$ can be approximated within error $\epsilon_1 + \epsilon_2$ by a ReLU network with depth at most $\max\{d_1, d_2\} + 2$ and size at most $s_1 + s_2 + 2$.*

*Proof.* We represent $I_{\mathcal{M}_1 \cup \mathcal{M}_2} = I_{x>0} \circ (I_{\mathcal{M}_1} + I_{\mathcal{M}_2})$ by a threshold function $I_{x>0}$. The threshold function can be approximate by a function

$$f(x) = \begin{cases} 0, & x \leq 0 \\ \frac{x}{\delta}, & x \in (0, \delta) \\ 1, & x \geq \delta. \end{cases} \tag{A.24}$$

with errors only in $(0, \delta)$. $f$ can be represented by a 2-layer ReLU network $h_f$ with size 2. Then if let $h = h_f \circ (h_1 + h_2)$, according to Lemma 4 and 3, $h$ is a neural network with depth $\max\{d_1, d_2\} + 2$ and size $s_1 + s_2 + 2$. Then we bound the error

$$\begin{aligned} ||h - I_{\mathcal{M}_1 \cup \mathcal{M}_2}||_{L_2(\mu)} &= ||h_f \circ (h_1 + h_2) - I_{x>0} \circ (I_{\mathcal{M}_1} + I_{\mathcal{M}_2})||_{L_2(\mu)} \\ &\leq ||h_f \circ (h_1 + h_2) - f \circ (I_{\mathcal{M}_1} + I_{\mathcal{M}_2})||_{L_2(\mu)} \\ &\quad + ||f \circ (I_{\mathcal{M}_1} + I_{\mathcal{M}_2}) - I_{x>0} \circ (I_{\mathcal{M}_1} + I_{\mathcal{M}_2})||_{L_2(\mu)} \\ &\leq ||h_f \circ (h_1 + h_2) - h_f \circ (I_{\mathcal{M}_1} + I_{\mathcal{M}_2})||_{L_2(\mu)} \\ &\leq ||h_f||_\infty ||(h_1 + h_2) - (I_{\mathcal{M}_1} + I_{\mathcal{M}_2})||_{L_2(\mu)} \\ &\leq ||(h_1 + h_2) - (I_{\mathcal{M}_1} + I_{\mathcal{M}_2})||_{L_2(\mu)} \\ &\leq \epsilon_1 + \epsilon_2 \end{aligned} \tag{A.25}$$

$\square$

**Theorem 1** (Complexity Arising from Topology)**.** *Suppose $\mathcal{M}'$ is the topological representative of d-manifold from the thickened 1-manifold family. Given $\epsilon > 0$, there exists a ReLU network $h : \mathbb{R}^d \to \mathbb{R}$ with depth at most $O(\log \beta)$ and size at most $O(\frac{d^2 \beta^2}{\epsilon})$, that can approximate the indicator function $I_{\mathcal{M}'}$ with error $R(h) \leq \epsilon$ for any continuous distribution $\mu$ over $\mathbb{R}^d$. $\beta$ is the topological complexity of $\mathcal{M}'$.*

*Proof.* Since $\mathcal{M}'$ is a topological representative, according to Lemma 1, there exist a set of $m_1$ $d$-balls $\mathcal{B} = \{B_{r_i}^d(\mathbf{c}_i)\}_{i=1}^{m_1}$ and a set of $m_2$ solid $d$-tori $\mathcal{T} = \{T_{r_i, R_i}^d(\mathbf{c}_i)\}_{i=1}^{m_2}$, such that $\mathcal{M}' = (\bigcup_{B \in \mathcal{B}} B) \cup (\bigcup_{T \in \mathcal{T}} T)$. Let $m = m_1 + m_2$. According to Lemma 7, $I_{\mathcal{M}'}$ can be approximated by a ReLU network $h$ with depth at most $\max\{d_1, d_2, ..., d_m\} + \log m$ and size at most $\sum_{i=1}^m s_i + \log m$, with error $R(h) \leq \sum_{i=1}^m \epsilon_i$. Then according to Proposition 1 and 2, $s_i \sim O(d^2/\epsilon_i)$, $d_i \sim O(1)$ and take $\epsilon_i$ to be all the same for all $i = [m]$. Let $\epsilon = m\epsilon_i$ and note that $m \leq \beta$. We have $h$ has depth at most $O(\log \beta)$ and size at most $O(\frac{d^2 \beta^2}{\epsilon})$, and can approximate $I_{\mathcal{M}'}$ with error $R(h) \leq \epsilon$. $\square$

## A.5 Overall Complexity

We present a result from Gonzalez-Diaz et al. (2019), which gives a bound of network size to represent a simplicial map.

**Proposition 6** (Adapted from Theorem 4 in Gonzalez-Diaz et al. (2019))**.** *Let us consider a simplicial map $\phi_c : |K| \to |L|$ between the underlying space of two finite pure simplicial complexes $K$ and $L$. Then a two-hidden-layer feed-forward network $\mathcal{N}_\phi$ such that $\phi_c(x) = \mathcal{N}_\phi(x)$ for all $x \in |K|$ can be explicitly defined. The size of $N_f$ is $D + d + k(D+1) + l(d+1)$, where $D = dim(|K|)$ and $d = dim(|L|)$, $k$ and $l$ are the number of simplices in $K$ and $L$, respectively.*

**Theorem 2** (Main Theorem)**.** *Let $\mathcal{M} = \mathcal{M}_1 \sqcup \mathcal{M}_0 \subset \mathbb{R}^D$ be a d-dimensional manifold from the thickened 1-manifold family. $\mathcal{M}_1$ and $\mathcal{M}_0$ are two disjoint sub-manifolds of $\mathcal{M}$ representing two classes. The condition number of $\mathcal{M}$ is $\frac{1}{\tau}$ and the total Betti number of $\mathcal{M}_1$ is $\beta$. Given a training set $\{(\mathbf{x}_i, y_i) | \mathbf{x}_i \in \mathcal{M}, y_i \in \{0, 1\}\}_{i=1}^n$, where $\mathbf{x}_i$ are sampled i.i.d. from $\mathcal{M}$ by a uniform distribution, and $y_i = I_{\mathcal{M}_1}(\mathbf{x}_i)$. For any $\delta > 0$, if inequality (8) holds, then for any $\epsilon > 0$, there exists a ReLU network $g$ with depth at most $O(\log \beta + d \log \frac{1}{\tau} + \log \log \frac{1}{\tau \delta})$ and size at most $O(\frac{d^2 \beta^2}{\epsilon} + \tau^{-d^2/2} \log^{d/2} \frac{1}{\tau \delta} + D\tau^{-d} \log \frac{1}{\tau \delta})$, such that*

$$P(R(g) \leq \epsilon) > 1 - \delta, \tag{A.26}$$

*where $R(g) = \int_{\mathcal{M}} (g - I_{\mathcal{M}_1})^2 \mu(\mathbf{x}) d\mathbf{x}$ with any continuous distribution $\mu$.*

*Proof.* Since $\mathcal{M} = \mathcal{M}_1 \sqcup \mathcal{M}_0$ is from thickened 1-manifold family, it has a topological representative $\mathcal{M}' = \mathcal{M}'_1 \sqcup \mathcal{M}'_0 \subset \mathbb{R}^d$, where $\mathcal{M}'_1$ and $\mathcal{M}'_0$ are topological representatives of $\mathcal{M}_1$ and $\mathcal{M}_2$, respectively.

The proof follows by first constructing simplicial approximations $|K|$ and $|L|$ of $\mathcal{M}$ and $\mathcal{M}'$, respectively. Then we represent a simplicial homeomorphism $\phi : |K| \to |L|$ by a neural network $N_\phi$, where $K$ is constructed from $\mathcal{M}$ and $L$ from $\mathcal{M}'$. Built on the top of this, a projection from $\mathcal{M}$ to its simiplicial approximation $|K|$ is represented by another network $N_p$. The overall network can be constructed by $g = h \circ N_\phi \circ N_p$. Note that $h$ is the function to approximate $I_{\mathcal{M}'_1}$, but the data after projection and homeomorphism is from $|L_1|$. There should be an error in this approximation. However, we will show that by using the true risk, having $|L_1| \subseteq \mathcal{M}'_1$ will make sure $||I_{|L_1|} - I_{\mathcal{M}'_1}||_{L_2(\mu')} = 0$. We move ahead by first constructing the network $g$, and then bounding the approximation error.

**Network Construction.** Given $\mathcal{M}$ is a compact submanifold of $\mathbb{R}^D$ and $\mathbf{x}_i$ are sampled according to a uniform distribution, by Proposition 3, for all $0 < r < \tau/2$ and $n > \lambda_1(log(\lambda_2) + log(\frac{1}{\delta}))$ $(n \sim O(\tau^{-d} \log(1/\tau\delta)))$, $U = \bigcup_i B_r^D(\mathbf{x}_i)$ has the same homology as $\mathcal{M}$ with probability higher than $1 - \delta$. Note that every $B_r^D(\mathbf{x}_i)$ is contractible because $r \leq \tau$. Therefore by the nerve theorem (Edelsbrunner & Harer, 2022), the nerve of $U$ is homotopy equivalent to $\mathcal{M}$. Note that $U$ is a collection of $\epsilon$-balls. The nerve of $U$ is the Čech complex, which is an abstract complex constructed as $\text{Čech}(r) = \{\sigma \subseteq X | \bigcap_{\mathbf{x} \in \sigma} B_r(\mathbf{x}) \neq 0\}$. But since the dimension of $\mathcal{M}$ is $d$, it suffices to only consider simplices with dimension $\leq d$. Delaunay complex is such a geometric construction that limits the dimension of simplices we get from a nerve. And in the other hand, we also do not want to lose the radius constraint. Here we construct the Alpha complex, a sub-complex of the Delaunay complex. It is constructed by intersecting each ball with the corresponding Voronoi cell, $R_\mathbf{x}(r) = B_r(\mathbf{x}) \cap V_\mathbf{x}$. The alpha complex is defined by

$$\text{Alpha}(r) = \{\sigma \subseteq X | \bigcap_{\mathbf{x} \in \sigma} R_\mathbf{x}(r) \neq 0\}. \tag{A.27}$$

Based on the construction, $\text{Alpha}(r)$ also has the same homotopy type as $U$. Bern et al. (1995) provided the number of simplices in a Delaunay complex of $n$ vertices is bounded by $O(n^{\lceil d/2 \rceil})$. Since the Alpha complex is a sub-complex of Delaunay complex, the number of simplices in $\text{Alpha}(r)$ is also bounded by

$$O(n^{\lceil d/2 \rceil}) = O(\tau^{-d^2/2} \log^{d/2} \frac{1}{\tau\delta}) \tag{A.28}$$

Denote $K = \text{Alpha}(r)$.

We claim that there exists a a vertex map $\phi : \mathbf{x}_i \to \mathbf{x}'_i$ for $i = 1, ..., n$, such that with probability higher than $1 - \delta$, $U' = \bigcup_i B_{r'}^d(\mathbf{x}'_i)$ has the same homology of $\mathcal{M}$. We prove this claim after the proof. We can construct an alpha complex from $\{\mathbf{x}'_i\}_{i=1}^n$ in a similar way, $L = \text{Alpha}(r)$. The number of simiplices is also bounded by $O(\tau^{-d^2/2} \log^{d/2} \frac{1}{\tau\delta})$.

$\phi$ can be extended to a simiplicial map $\phi : |K| \to |L|$ by

$$\phi(\mathbf{x}) = \sum_{i=1}^n b_i(\mathbf{x})\phi(\mathbf{x}_i). \tag{A.29}$$

The map $b_i : |K| \to \mathbb{R}$ maps each point to its $i$-th barycentric coordinate. According to Proposition 6, there exists a ReLU network $N_\phi$ with depth 4 and size $O(\tau^{-d^2/2} \log^{d/2} \frac{1}{\tau\delta})$, such that $\phi(\mathbf{x}) = N_\phi(\mathbf{x})$ for all $\mathbf{x} \in |K|$.

Next we construct a network $N_p$ that projects $\mathcal{M}$ to its simplicial approximation $|K|$. The point is projecting $\mathbf{x} \in \mathcal{M}$ to its closest simplex $\sigma_\mathbf{x}$. According to the proof of theorem 3 in Schonsheck et al. (2019), such projection can be represented as a neural network $N_p$ with depth at most $\log n + 1$ and size at most $O(nD)$. Lastly, by Theorem 1, a neural network $h$ with depth at most $O(\log \beta)$ and size at most $O(\frac{d^2\beta^2}{\epsilon_1})$ can approximate $I_{\mathcal{M}'_1}$ with error $R(h) \leq \epsilon_1$. And by Lemma 3, $g = h \circ N_\phi \circ N_p$ has depth at most $O(\log(n\beta))$ and size $O(\frac{d^2\beta^2}{\epsilon_1} + \tau^{-d^2/2} \log^{d/2} \frac{1}{\tau\delta} + nD)$. Given $n \sim O(\tau^{-d} \log(1/\tau\delta))$, $g$ has depth at most $O(\log \beta + d \log \frac{1}{\tau} + \log \log \frac{1}{\tau\delta})$ and size at most $O(\frac{d^2\beta^2}{\epsilon_1} + \tau^{-d^2/2} \log^{d/2} \frac{1}{\tau\delta} + D\tau^{-d} \log \frac{1}{\tau\delta})$. Note that the probability of the existence for such network is larger than $(1 - \delta)^2 = 1 - 2\delta + \delta^2 > 1 - 2\delta$. We let

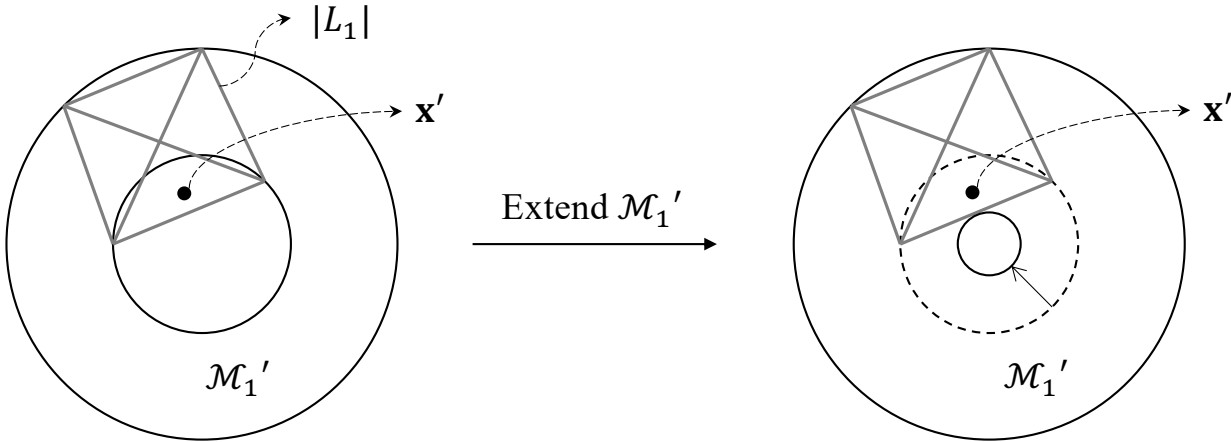

Figure A.2: For a point $\mathbf{x}' \in |L_1|$ but $\mathbf{x}' \notin \mathcal{M}_1'$, we extend $\mathcal{M}_1'$ to make $\mathbf{x}' \in \mathcal{M}_1'$. The extended $\mathcal{M}_1'$ is still a topological representative.

$\delta = 2\delta$, such that with probability larger than $1 - \delta$, neural network $g$ exists and $g$ has depth at most $O(\log \beta + d \log \frac{1}{\tau} + \log \log \frac{1}{\tau\delta})$ and size at most $O(\frac{d^2\beta^2}{\epsilon_1} + \tau^{-d^2/2} \log^{d/2} \frac{1}{\tau\delta} + D\tau^{-d} \log \frac{1}{\tau\delta})$.

**Bounding Approximation Error.** Now it is time to bound the approximation error $R(g)$. We split $R(g)$ into two parts.

$$
\begin{aligned}
R(g) &= ||g - I_{\mathcal{M}_1}||_{L_2(\mu)} \\
&\leq ||h \circ N_\phi \circ N_p - I_{\mathcal{M}_1'} \circ \phi \circ N_p||_{L_2(\mu)} + ||I_{\mathcal{M}_1'} \circ \phi \circ N_p - I_{\mathcal{M}_1}||_{L_2(\mu)} \\
&= I_1 + I_2.
\end{aligned}
\tag{A.30}
$$

We first show that $I_2 = 0$. Note that $N_p : \mathcal{M} \to |K|$, and $\phi : |K| \to |L|$. We claim that for $\mathbf{x} \in \mathcal{M}_1$, $\phi \circ N_p(\mathbf{x}) \in L_1$ and if $\mathbf{x} \in \mathcal{M}_0$, $\phi \circ N_p(\mathbf{x}) \in L_0$. This is true because $\mathcal{M}_1$ and $\mathcal{M}_2$ are disjoint and $L$ is homotopy equivalent to $\mathcal{M}$. Consequently, $I_{L_1} \circ \phi \circ N_p = I_{\mathcal{M}_1}$. Given

$$
\begin{aligned}
||I_{\mathcal{M}_1'} \circ \phi \circ N_p - I_{\mathcal{M}_1}||_{L_2(\mu)} &= ||I_{\mathcal{M}_1'} \circ \phi \circ N_p - I_{L_1} \circ \phi \circ N_p||_{L_2(\mu)} \\
&= ||I_{L_1} - I_{\mathcal{M}_1'}||_{L_2(\mu')}.
\end{aligned}
\tag{A.31}
$$

The second equation is derived by letting $\mu'(\mathbf{x}') = \mu \circ \phi \circ N_p(\mathbf{x})$. Now it suffices to show $||I_{L_1} - I_{\mathcal{M}_1'}||_{L_2(\mu')} = 0$. Note that $\mu'$ is a distribution supported on $|L|$, it can be naturally extended to $\mathcal{M}'$ by set $\mu'(\mathbf{x}') = 0$ if $\mathbf{x}' \in \mathcal{M}'$ but $\mathbf{x}' \notin |L|$. For $\mathbf{x}' \in |L_1|$ but $\mathbf{x}' \notin \mathcal{M}_1'$, like shown in Figure A.2, we extend $\mathcal{M}_1'$ so that $\mathbf{x}' \in \mathcal{M}_1'$. And such extension always exists due to the construction of $|L_1|$. Note that $|L_1|$ is a alpha complex constructed from a $r'$-cover, in a way that there will be an edge if and only if two covering balls have intersection. Hence, for any edge $(\mathbf{x}_i', \mathbf{x}_j') \in |L_1|$, the length $l_{ij}$ of it satisfies $l_{ij} < 2r' < \tau'$. And notice the radius of the inner circle should be at least $\tau'$. Otherwise, the reach will be less than $\tau'$. Consequently, the edge of $|L_1|$ is always smaller than the radius of the inner circle. Therefore, one can always choose a $\mathcal{M}_1'$, so that $\forall \mathbf{x}' \in |L_1|$ but $\mathbf{x}' \in \mathcal{M}_1'$.

After the expansion, $|L_1| \subseteq \mathcal{M}_1'$. As a conclusion, $||I_{L_1} - I_{\mathcal{M}_1'}||_{L_2(\mu')} = 0$. Hereby, we have proved $I_2 = 0$.

Now we settle $I_1$. Given $N_\phi$ is an exact representation of the simplicial map $\phi$,

$$
I_1 = ||h - I_{\mathcal{M}_1'}||_{L_2(\mu')}.
\tag{A.32}
$$

According to Theorem 1, $I_1 \leq \epsilon_1$. Combined together, we have

$$
R(g) \leq \epsilon_1.
\tag{A.33}
$$

Note that this inequality holds only with probability larger than $1 - \delta$ because that is the probability we successfully recover the homology of $\mathcal{M}$ by the training set and construct a simplicial homeomorphism. $\square$

**Claim 1.** $\mathcal{M} \in \mathbb{R}^D$ *is a d-dimensional manifold from thickened* $1$*-manifold family. Suppose there exists a set* $\{\mathbf{x}_i \in \mathcal{M}\}_{i=1}^n$ *and radius* $r$*, such that* $U = \bigcup_i B_r^D(\mathbf{x}_i)$ *is a cover of* $\mathcal{M}$ *and has the same homology. Then there exists a* $\mathcal{M}' \in \mathbb{R}^d$ *that is a topological representative of* $\mathcal{M}$*. Denote the homeomorphism between them* $f$*. Then with probability larger than* $1 - \delta$*,* $U' = \bigcup_i B_{r'}^d(f(\mathbf{x}_i))$ *has the same homology as* $\mathcal{M}$*.*

*Proof.* We let

$$c(r, \tau, \mathcal{M}) = \frac{vol(\mathcal{M})}{(cos^d\theta_1)vol(B_{r/4}^d)} \left( \log \frac{vol(\mathcal{M})}{(cos^d\theta_2)vol(B_{r/8}^d)} + \log \frac{1}{\delta} \right), \tag{A.34}$$

where $\theta_1 = \arcsin \frac{r}{8\tau}$, $\theta_2 = \arcsin \frac{r}{16\tau}$ and $0 < r < \tau/2$. Given a set $\{f(\mathbf{x}_i)\}_{i=1}^n$, apply proposition 3 to $\mathcal{M}'$. If

$$n > c(r', \tau', \mathcal{M}'), \tag{A.35}$$

then with probability $1 - \delta$, $U' = \bigcup_i B_{r'}^d(f(\mathbf{x}_i))$ has the same homology as $\mathcal{M}'$, with $r' < \tau'/2$.

Note that $n$ already satisfy that $n > c(r, \tau, \mathcal{M})$, it suffices to show $c(r, \tau, \mathcal{M}) > c(r', \tau', \mathcal{M}')$. Since $\mathcal{M}'$ is one of topological representatives of $\mathcal{M}$, we can always choose the radius of the fundamental members in $\mathcal{M}'$ and choose the distance between $\mathcal{M}'_1$ and $\mathcal{M}'_2$, to make sure that $\tau' > \tau$ and $vol(\mathcal{M}') < vol(\mathcal{M})$. Hence, we can choose $r$ and $r'$, such that $B_{r'}^d > B_r^d$. With the same $\delta$, we have proved that $c(r, \tau, \mathcal{M}) > c(r', \tau', \mathcal{M}')$.

$\square$

