# OpenReview forum: "A Theoretical Study of Neural Network Expressive Power via Manifold Topology"
_TMLR — Accepted by TMLR_

### Review · Reviewer_KGYi · 2025-02-08

**Summary Of Contributions:**

This is a theoretical work that proposes a new way of analyzing the relationship between the complexity of a data manifold and the capacity required for a neural network to perform classification on it. The authors focus on binary classification for this analysis.

The authors point out that past work in this vein has focused on geometric characteristics (especially local intrinsic dimension/LID) but not topological characteristics of the manifolds. The key theme and contribution of the work is about separating the geometric properties (here characterized by the condition number) and the topological properties (here characterized by the total Betti number $\beta$).

This separation is achieved as follows:
- First, a ReLU network is constructed to approximate a homeomorphism that maps the manifold into a union of $d$-balls and solid $d$-tori. The size of this network scales with the condition number (geometric complexity). This homeomorphism "removes" the geometric complexity of the data manifold, making it easy to characterize its topological complexity in the next step.
- Second, another ReLU network is constructed on the codomain of the first which approximates the ground truth classifier and whose size scales with $\beta$ (the topological complexity).

The authors provide a small experiment showing that classification on a synthetic manifold actually does require network size linear in $\beta^2$, as predicted by their theory.

**Audience:**

Yes

**Claims And Evidence:**

Yes

**Requested Changes:**

Typos
- p5 "Reach and *conditional* number"
- p8 "*simiplicial* approximation"
- p18 "proposition 7" in the proof of Theorem 1 should read "Lemma 7"

Not necessary for acceptance: Can you elaborate further on why the thickened 1-manifold family should be considered representative of true data manifolds?

**Strengths And Weaknesses:**

In my opinion, this is a high quality theory paper.
- As supported by the experiment in section 5, this is highly relevant for the field and for practical neural networks. How manifold structure enables learnability is one of the greatest mysteries in ML, and the authors are correct that global (topological) properties are particularly understudied in this respect.
- It is very well-written with a 5/5 for clarity. Even though its content is esoteric, the authors have clearly put a great deal of care into making it accessible with clear, intuitive prose and useful diagrams. Their ideas are well-distilled into key themes which are repeatedly emphasized throughout the paper.

I did not read the proofs in the appendix in detail and cannot vouch for their correctness.

The only weakness I can point out is their strong assumptions on the data manifold. They authors write that "this family may appear overly restrictive; however, we will explain why it actually represents a broad and versatile class of manifolds." I did not find any such explanation in the main text.

---

> ### Author Response · Authors · 2025-02-16
> **Author Reply to Reviewer KGYi**
>
> We sincerely appreciate your thoughtful and constructive review of our work. Your positive assessment of our theoretical contributions and the clarity of our writing is truly encouraging. We are particularly grateful for your recognition of the relevance of our study to the broader question of how manifold structure influences learnability in neural networks.
>
> We acknowledge your concerns regarding the assumptions on the data manifold and believe they pertain to two key questions: (1) why the proposed manifold represents a broad and versatile class of manifolds, and (2) why it should be considered representative of real data manifolds. We address these concerns as follows.
>
> First, when we state that “this family may appear overly restrictive; however, we will explain why it actually represents a broad and versatile class of manifolds,” we emphasize that the proposed thickened 1-manifold family goes beyond standard examples like cylinders and solid tori. It also includes any manifold that is homeomorphic to such thickened shapes. For instance, a cube in $\mathbb{R}^3$ can be continuously deformed (retracted) to a 1-dimensional skeleton, effectively showing it is homeomorphic to a 1-manifold (a point or curve) thickened by a disk. As illustrated in Figure 2 for a simpler 2D setting, 2-manifolds with boundaries can likewise be described via a thickened 1-dimensional core.
>
> Second, while real-world data, such as images, are believed to reside on manifolds of much lower intrinsic dimension, their precise properties remain unknown and are often impossible to determine. To make the problem tractable, assumptions about the latent manifolds are typically introduced. Recent approaches, such as [1], have successfully learned thickened and deformed 1-manifolds from real-world data (e.g., MNIST) by leveraging the concept of principal curves[2]. Principal curves serve as a fundamental representation in which data points are assumed to be sampled from a probability distribution centered around these curves. Similarly, in probabilistic models such as kernel density estimation (KDE), each data point is treated as being distributed around a k-dimensional normal distribution, which effectively thickens a point into a k-dimensional ball. This geometric model aligns with our thickened 1-manifold framework, further supporting its applicability to real-world data.
>
> Additionally, we appreciate your careful reading and have corrected the noted typos, including “conditional number” (should be “condition number”), “simiplicial approximation” (should be “simplicial approximation”), and the reference to “proposition 7” (should be “Lemma 7” in Theorem 1’s proof). We have corrected these typos in the paper.
>
> We appreciate the opportunity to clarify this point and will incorporate a more detailed explanation in the revised version. Thank you again for your valuable feedback.
>
> [1] Cui, Elvis Han, and Sisi Shao. "A Metric-based Principal Curve Approach for Learning One-dimensional Manifold." arXiv preprint arXiv:2405.12390 (2024).
>
> [2] T. Hastie and W. Stuetzle. Principal curves. Journal of the American Statistical Association, 84(406):502–516, 1989.

---

> ### Comment · Reviewer_KGYi · 2025-02-22
> **Thanks for your reply**
>
> Thank you for addressing my question about the generality of 1-manifolds. The added context and explanation is helpful, but I don't find it completely satisfying. Though some data can be successfully modelled with thickened 1-manifold models such as principal curve approaches or KDE, the fact remains that these are poor models for the high-dimensional real-world data we deal with a deep learning context. Nevertheless, I encourage you to include further discussion of this point in the next version of your manuscript.

---

### Review · Reviewer_98WJ · 2025-02-13

**Summary Of Contributions:**

The manuscript presents an upper bound on the size of a ReLU network required to achieve a given error level in binary classification on data from a manifold representing two classes. The manifold is assumed to be a finite combination of thickened 1d manifolds (made of balls around all points in a line or a circle), and the bound is given in terms of its geometrical and topological properties, namely condition number (related to curvature) for geometric complexity and Betti numbers for topological complexity. The manuscript decouples the two properties by mapping the manifold to the latent space, using previous results on the network size needed for that given the geometric complexity, and then using the topologic complexity to analyse the needed network size to achieve the target error in that space. The scaling predicted by topology can be observed in simulations on synthetic data.

**Audience:**

Yes

**Claims And Evidence:**

No

**Requested Changes:**

* Is it possible to demonstrate in simulations how changes to the geometry of manifolds interfere with changes in topology? For example, repeat the experiment reported in Figure 4c on manifolds with the same dimension but higher or lower curvature. Then see if the interaction observed is in line with what is predicted by the results of the manuscript.
 * Clarify if the results in the paper should be interpreted as scaling laws or as upper bounds; if they are upper bounds, show how to apply them to results (in terms of the free parameters).
 * Missing references to manifold classification literature (e.g., https://doi.org/10.1103/PhysRevX.8.031003) and discussion of the different approaches.

**Strengths And Weaknesses:**

Strengths:
 * Data topology may provide a different perspective on the ability to perform classification, which is not captured in current methods, which mostly rely on geometry.
 * An interesting upper bound decouples the contributions of Betti numbers and curvature to the network's size.

Weaknesses
 * The result seems to apply only to thickened 1d manifolds. It is unclear if it can be extended to more complicated manifolds or if more complicated ones can be somehow reduced to those simple ones.
 * The result can be interpreted as an upper bound, which may be vacuous, or as a scaling argument on the ability to perform classification. The supporting evidence in simulations is only for the first half, i.e., Theorem 1, laying out the contribution of topology. However, no support is provided for the second half, i.e., Theorem 2, combining the contributions of topology and geometry. The authors state Theorem 2 as their main contribution.
 * The decoupling argument used in Theorem 2 seems naive to yield a non-vacuous upper bound and should be rephrased as a naive upper bound or better supported by simulations.

---

> ### Author Response · Authors · 2025-02-27
> **Author Reply to Reviewer 98WJ**
>
> We would like to thank you for your constructive and thorough feedback on our manuscript. Your insights and suggestions have been invaluable, and we appreciate the opportunity to address them in detail. Below, we respond to each of your main comments:
> Generality of the proposed manifold: We have revised the manuscript to better clarify the thickened 1-manifold family. Please refer to the first point in the summary response for more details.
>
> 1. **The nature of the bound:** Strictly speaking, our result is an upper bound on the network size needed to achieve a given classification error rate, rather than a scaling law. While it offers scaling insights into how geometric and topological complexities influence network size, it does not necessarily pinpoint the minimum possible size. Empirical evaluations suggest that the topological part of our bound can be tight, whereas the geometric part tends to be more conservative, since we did not adapt it to incorporate improvements from prior work [1, 2]. Nonetheless, the bound is not vacuous. It underscores how the geometry of data manifolds can affect network size in an exponential manner, in contrast to the quadratic scaling from topological complexity. Moreover, even a relatively loose upper bound can still be practically sufficient for deciding network size, as overparameterization often leads to improved generalization [3].
>
> 2. **How changes to the geometry of manifolds interfere with changes in topology:** Thank you for this insightful suggestion. We have added an experiment in the appendix where we generate a torus with a smaller inner radius ($r$), thereby reducing its reach. Please refer to section 5 in the updated manuscript and the second point in the summary response for more details.
>
> 3. **References:** Thank you for recommending a relevant paper. We have incorporated a discussion of it in our related work section.
>
> [1] Stefan C. Schonsheck, Jie Chen, and Rongjie Lai. Chart auto-encoders for manifold structured data. CoRR, abs/1912.10094, 2019.
>
> [2] Minshuo Chen, Haoming Jiang, Wenjing Liao, and Tuo Zhao. Efficient approximation of deep relu networks for functions on low dimensional manifolds. In Advances in Neural Information Processing Systems, 2019.
>
> [3] Allen-Zhu Z, Li Y, Liang Y. Learning and generalization in overparameterized neural networks, going beyond two layers. Advances in neural information processing systems. 2019;32.

---

### Review · Reviewer_hzhS · 2025-02-16

**Summary Of Contributions:**

The current paper explored how a classifier network size scales up with respect to the topological complexity (given by the Betti numbers) and curvature of the data manifold (given by the the reach) via a constructive proof.

**Audience:**

Yes

**Broader Impact Concerns:**

NA.

**Claims And Evidence:**

Yes

**Requested Changes:**

Please argue carefully the reasonableness of using the thickened 1-manifold to model real data.

Please explain in detail how to compute the Betti numbers in practice when the data is high dimensional and sparse.

Related work should be more carefully discussed as there are existing work on applying topological properties to neural network expressiveness.

**Strengths And Weaknesses:**

Pros:
The theoretical contribution is novel and interesting. The authors took a promising direction on incorporating the topological complexity of data.

Cons:
The validity of modeling real data as thickened 1-manifolds needs more justification (as a separate section).

The practical applicability of computing the proposed bounds on real datasets needs more verification (as a separate section). However the current empirical results are only on a toy dataset and are insufficient.

Due to these limitations, the paper needs more iterations before getting published. Moreover, the authors are encouraged to strengthen the empirical verification, and properly position this work in the literature of similar bounds.

---

> ### Author Response · Authors · 2025-02-27
> **Author Reply to Reviewer hzhS**
>
> Thank you for taking the time to review our manuscript and for acknowledging the novelty of our theoretical contribution. We appreciate your interest in how topological complexity and curvature can influence network size. Below, we respond to your comments and clarify our objectives:
>
> **1. Justification for Thickened 1-Manifolds**
>
> We argue that the proposed thickened 1-manifold family is highly applicable to real-world datasets. Data such as images are commonly assumed to reside on lower-dimensional manifolds. However, estimating the underlying manifold is challenging due to the limited sample size relative to the high ambient dimension. Despite this, assuming that the data lie on a thickened 1-manifold is not unreasonable. Please refer to the first point in the summary response for more details.
>
>
> **2. Computing Betti Numbers in Practice**
>
> Although computing manifold Betti numbers can be computationally expensive, our proposed thickened 1-manifold framework only requires the 0- and 1-dimensional Betti numbers. In practice, these are typically derived from the k-nearest neighbor (k-NN) graph, which is relatively efficient to construct. Specifically, after building a $k$-NN graph for a chosen $k$, a distance matrix $\delta_k(i, j)$ is computed as the number of edges between two points $i$ and $j$ when they are connected. A Vietoris--Rips complex $VR_{k,\epsilon}$ is then formed with respect to the metric $\delta_k$ and a chosen scale $\epsilon$, and the Betti numbers of $VR_{k,\epsilon}$ approximate the Betti numbers of the underlying data manifold. For instance, [3] provides examples of computing Betti numbers on both synthetic and real-world datasets, demonstrating the overall feasibility. Concretely, calculating the 0- and 1-dimensional Betti numbers for all 10 classes of MNIST (70,000 points in 784-dimensional space) takes roughly 9 seconds on a typical machine. Hence, we consider this computation practical, provided it is not repeated frequently during training.
>
> Nevertheless, in real-world scenarios, the underlying manifold is unknown, so any Betti numbers calculated are inherently approximations that depend on the sampling scale. In particular, sample sparsity can significantly constrain the accuracy of these estimates, making it difficult to fully capture the true manifold structure—a challenge not unique to topological methods, but also present in many machine learning techniques where limited or uneven data sampling hinders reliable inference.
>
> [3] Gregory Naitzat, Andrey Zhitnikov, and Lek-Heng Lim. “Topology of deep neural networks.” The Journal of Machine Learning Research, 21(1):7503–7542, 2020.
>
> **3. Positioning in the Literature**
>
> We carefully reviewed prior work on topological approaches to neural networks and expressiveness in section 2. If there are specific references we have not cited, we would be grateful if you could point them out. Otherwise, we will re-check and further clarify how our work relates to existing results on topological complexity, ensuring readers see the connections and differences clearly.

---

### Review · Reviewer_RNER · 2025-02-25

**Summary Of Contributions:**

The paper studies how the minimum size of a ReLU network necessary to express the solution to a binary classification problem on a topological manifold changes as a function of the topological complexity (quantified by the sum of the Betti numbers). More precisely, the author's present a novel method for disentangling the impact of geometric complexity (quantified by the condition number of a manifold) and topological complexity on bounds for network size by:
- considering a restricted family of manifolds (thickened 1-D manifolds),
- constructing an architecture that first learns a homeomorphism between an input manifold and a "topological representative." The size of a network needed to implement this function is  governed by the geometric complexity.
-  proving that the required size for solving the classification problem on the topological representative is bounded by a term quadratic in the total Betti number.

The author's conduct an experiment to demonstrate that the relationship between the minimum network size and the square of the total Betti number (of an input dataset) is indeed linear (as predicted by the theory) in a simple setting.

**Audience:**

Yes

**Claims And Evidence:**

Yes

**Requested Changes:**

In relation to the weaknesses mentioned above:
- Could the author's add some more explicit examples of interesting manifolds in the thickened 1-manifold family? I appreciate that the connected sum and disjoint union operations make the family more expressive than one might appreciate at first glance, but think this could be better presented in the main text (I think figure A.1 could even be promoted to the main text). It would also be good to discuss what types of manifolds lie outside this family.
- Can the author's explicitly acknowledge that the derived bound need not be tight (i.e., the relationship should not be thought of as a scaling law)? It would be particularly interesting if one could design a simple input set where you expect the relationship between network size and the square of the Betti number to be sublinear. Perhaps this really only makes sense to note in the discussion section?
- If possible, it would be nice to expand the empirical evaluations. Ideally I think at least two types of input manifolds (which would each be distinct types of topological representatives) and varying dimensionality  should be considered. If the linearity observed in Fig 4c is preserved that would be quite interesting!

Typo: in the proof sketch at the bottom of page 8:  "...$\mathcal{M}'_1$ and $\mathcal{M}'_0$ are topological representatives of $\mathcal{M}_1$ and $\mathcal{M}_2$...", should be $\mathcal{M}_1$ and $\mathcal{M}_0$.

None of the changes are required to recommend acceptance, but I do believe each would strengthen the work.

**Strengths And Weaknesses:**

Strengths:
- The idea of considering how the topological complexity of inputs impacts the necessary expressivity of NNs operating on such data is novel and interesting.
- The presentation is clear, with appropriate amounts of detail abstracted away/relegated to appendices and the remaining formalisms well described. This was really a non-trivial thing to do given the nature of the content and I applaud the author's for their efforts.

Weaknesses:
- Analysis is restricted to the family of thickened 1-manifold family: While the structure of this family facilitates the proof by construction the paper would be strengthened by giving some examples of the types of properties that can and cannot be modeled with or approximated by this family.
- As far as I could tell, there are no arguments made that the derived bounds will be tight (though the empirics suggest this may be the case in a very simple setting).
- Empirical evaluation is limited: experiments are limited to tori of different genera (so $\beta$ varies) with a fixed dimension.

---

> ### Author Response · Authors · 2025-02-27
> **Response to Reviewer RNER**
>
> We thank Reviewer for their thoughtful and detailed review of our paper. We greatly appreciate the time and effort taken to engage deeply with our work and for the constructive feedback provided. Below, we address each of the reviewer’s comments and suggestions.
>
> **1. Examples of Manifolds in the Thickened 1-Manifold Family**
>
> Thank you for suggesting additional examples to illustrate the thickened 1-manifold family. We have revised the manuscript to further clarify this concept and provide more representative examples. Please see page 4 in the updated manuscript, as well as the first point in the summary response for details.
>
> **2. Clarification on the bound.**
>
> We appreciate this insightful point and agree that it is essential to clearly interpret the derived bounds. In the revised manuscript, we now explicitly state that the bounds represent upper limits and may not always be tight. Moreover, even if the bound were tight in some cases, this does not imply a scaling law, as a tight lower bound would also be required for such a claim. Nevertheless, this bound provides valuable insights into the different factors that contribute to network complexity.
>
> **3. More empirical evaluation.**
>
> We fully agree that expanding the empirical evaluation strengthens the paper. In response, we have added a new experiment to investigate how the geometric properties of the manifold influence the required neural network size. Please refer to Section 5 in the updated manuscript and the second point in the summary response for further details.
>
> We are grateful for your constructive feedback, which has helped us refine and improve our manuscript. We believe these revisions will enhance the clarity and impact of our work. Thank you again for your time and thoughtful review.

---

### Author Response · Authors · 2025-02-27
**Summary Response to All Reviewers**

We sincerely appreciate the reviewers’ valuable insights and their **recognition of the novelty of our theoretical contributions** as well as the effort we have put into **ensuring clarity**. While we include illustrative experiments, we would like to clarify that the **primary contribution** of this paper is **theoretical**. Based on your suggestions, we have carefully revised our manuscript, with major changes highlighted in **blue**. Below, we summarize the key modifications:

**1. Justification for thickened 1-manifold family.**

The proposed thickened 1-manifold family may seem straightforward at first glance. However, homeomorphisms, connected sums, and disjoint unions significantly enhance its expressivity, enabling a broad range of topological constructions. To emphasize this point, we have added a dedicated paragraph on page 4 and included more examples of the thickened 1-manifold family.
Notably, all manifolds that can be contracted into skeletons belong to the thickened 1-manifold family. Recent approaches, such as [1], have successfully learned thickened and deformed 1-manifolds from real-world data (e.g., MNIST), building on the concept principal curves [2]. Principal curves serve as a foundation where data are assumed to be sampled from a probability distribution centered around these curves. This aligns well with our characterization of the thickened 1-manifold family.

[1] Cui, Elvis Han, and Sisi Shao. "A Metric-based Principal Curve Approach for Learning One-dimensional Manifold." arXiv preprint arXiv:2405.12390 (2024).

[2] T. Hastie and W. Stuetzle. Principal curves. Journal of the American Statistical Association, 84(406):502–516, 1989.

**2. New Experiment: Impact of Geometric Complexity**

To examine whether geometric complexity affects the topological component of network expressivity, we have added a new experiment in Section 5. Specifically, we train a 5-layer neural network on tori with genus 1 to 10, but with a smaller inner radius, which corresponds to a larger condition number.
The results in Figure 6 continue to indicate a linear relationship between network size and $\beta^2 $. However, when compared to Figure 5c, the required network size is significantly larger for the same genus, highlighting the impact of geometric complexity on network expressivity.

 We hope that our work will spur future studies, both refining these theoretical bounds and exploring how to adapt such manifold-based insights more seamlessly to real-world datasets. Thank you again for your thoughtful feedback. We look forward to further discussions and potential extensions of this work.

---

### Decision · Action_Editor_4xqk · 2025-03-16

**Recommendation:** Accept with minor revision

**Comment:**

After the rebuttal, three reviewers agree it now exceeds the bar for admission, while one reviewer provided more thoughtful critiques.

AE has carefully considered all raised concerns, and wanted to share why he believes this manuscript merits acceptance despite proposed reservations:

1. Scope and Nature of the Paper

The primary contribution of this work is theoretical rather than empirical. While the authors do offer illustrative experiments (and do plan to add additional clarifications/experiments in revision), their goal is to build new theoretical insight on how topological characteristics of data manifolds can affect the capacity requirements of neural networks. Hence, although real-data experimental validations are an asset, the chief strength of the submission—and the main reason for my favorable view—lies in its theoretical contribution.

2. Reasonableness of the Thickened 1-Manifold Assumption

While concerns about the restrictiveness of a 'thickened 1-manifold' are understandable, the authors' definition—which includes disjoint unions, connected sums, and homeomorphisms—provides considerable expressive power. As they note, this family serves as a tractable approximation of real manifolds, capturing essential topological features (specifically the first two Betti numbers). Although it omits higher-dimensional topology—which is often computationally infeasible—it remains expressive enough to represent real-world data manifolds. Importantly, it lays the groundwork for a rigorous, topology-aware theoretical framework for understanding neural network expressive power.

3. Verification of Betti Number Computation in the revised manuscript

The point about verifying the stated 9s computation on MNIST is well-taken. We shall encourage the authors to clarify or add this computation time in their final version.

Based on these points, AE feels that the paper's claims are already supported by evidence and there is a relevant audience, hence meriting TMLR acceptance. AE will, of course, encourage the authors to explicitly address the specific criticisms (for instance, by including empirical evaluation of Betti number computations, and by discussing limitations in modeling real data as thickened 1-manifolds).

**Audience:**

Appropriate audience at TMLR

**Claims And Evidence:**

This paper presents a theoretical attempt towards understanding the relationship between learnability and data topology.